# CO2: Efficient Distributed Training with Full Communication-Computation Overlap

**Weigao Sun, Zhen Qin, Weixuan Sun, Shidi Li, Dong Li, Xuyang Shen,**
**Yu Qiao, Yiran Zhong**[*]

OpenNLPLab, Shanghai AI Laboratory
{sunweigao,zhongyiran}@pjlab.org.cn
https://github.com/OpenNLPLab/CO2

## Abstract

The fundamental success of large language models hinges upon the efficacious implementation of large-scale distributed training techniques. Nevertheless, building a vast, high-performance cluster featuring high-speed communication interconnectivity is prohibitively costly, and accessible only to prominent entities. In this work, we aim to lower this barrier and democratize large-scale training with limited bandwidth clusters. We propose a new approach called *CO2* that introduces local-updating and asynchronous communication to the distributed data-parallel training, thereby facilitating the full overlap of *CO*mmunication with *CO*mputation. CO2 is able to attain a high scalability even on extensive multi-node clusters constrained by very limited communication bandwidth. We further propose the staleness gap penalty and outer momentum clipping techniques together with CO2 to bolster its convergence and training stability. Besides, CO2 exhibits seamless integration with well-established ZeRO-series optimizers which mitigate memory consumption of model states with large model training. We also provide a mathematical proof of convergence, accompanied by the establishment of a stringent upper bound. Furthermore, we validate our findings through an extensive set of practical experiments encompassing a wide range of tasks in the fields of computer vision and natural language processing. These experiments serve to demonstrate the capabilities of CO2 in terms of convergence, generalization, and scalability when deployed across configurations comprising up to 128 A100 GPUs. The outcomes emphasize the outstanding capacity of CO2 to hugely improve scalability, no matter on clusters with 800Gbps RDMA or 80Gbps TCP/IP inter-node connections.

## 1 Introduction

Distributed optimization is crucial for the efficient training of large-scale deep neural networks. Mini-batch parallel optimization methods (Goyal et al., 2017; Li et al., 2014) like stochastic gradient decent (SGD) with distributed data parallel (DDP) paradigm are commonly used, but communication overhead can pose significant challenges when scaling out to larger GPU clusters. Existing techniques leverage gradient bucketing to partially overlap communication with backward computation to enhance training efficiency, but residual overhead remains a challenge in scenarios with large model sizes and limited inter-node communication bandwidth.

Various strategies have been proposed to address the communication-related issues. These strategies can be classified into three following categories: 1) *Communication Compression in Single Iteration*, which comprises techniques such as gradient sparsification (Shi et al., 2019; Li & Hoefler, 2022; Barnes et al., 2020) and gradient quantization (Alistarh et al., 2016; Tang et al., 2021; Li et al., 2021; Dettmers et al., 2021). These methods accelerate distributed training by minimizing the volume of communication traffic during each iteration. 2) *Communication Frequency Reduction* is represented by work such as (Lin et al., 2018; Wang et al., 2019; 2020; Wang & Joshi, 2021), which maintains the communication volume per iteration but lessens the frequency of communication events. 3) *Communication and Computation Overlapping* aims to overlap communication with computation,

---

[*]Corresponding author.

either in a single step or across multiple steps. For instance, the distributed module in PyTorch (Li et al., 2020) leverages gradient bucketing to partially overlap gradient all-reducing with the backward pass. Additionally, asynchronous distributed training methods (Dutta et al., 2021; Cohen et al., 2021; Mishchenko et al., 2022; Su et al., 2022; Koloskova et al., 2022) synchronize the transmission of stale gradients with the latest local computation on each worker.

These communication-efficient approaches improve convergence speed but often yield sub-optimal final optimization outcomes. None of these methods have achieved complete overlap of communication with computation, *i.e.,* 100% scalability throughout the entirety of the training process on a large cluster, especially under varying communication conditions while preserving the performance.

In this paper, we propose CO2 which enables complete overlap of *Co*mmunication with *Co*mputation. This is made possible through the use of the local updating strategy and asynchronous communication, where each worker node independently optimizes its model parameters without requiring synchronization with other peers at the local updating stage. As shown in Fig. 1, to guarantee parameter consistency across all workers, model parameter synchronization is performed asynchronously after completing every $\tau$ local updates. By selecting appropriate local updating step $\tau$ in accordance with the communication environment, we can achieve full overlap of model parameter synchronization with multiple local computation steps, thereby attaining 100% scalability.

Maintaining training stability as well as performance with asynchronous parameter updates is crucial for the success of our method. We introduce two additional techniques: staleness gap penalty and outer momentum clipping. The staleness gap penalty mechanism effectively quantifies and penalizes the discrepancy between different parameter versions during the update of outer momentum. This approach significantly contributes to the overall training stability and performance by addressing the inconsistencies in parameter updates. Similarly, outer momentum clipping serves as an essential tool to mitigate the emergence of anomalous values in the outer momentum. This helps prevent extreme values, thus further bolsters the training stability. In addition to our empirical findings, we establish robust theoretical convergence bounds. These bounds provide compelling evidence that our proposed framework is capable of achieving a convergence rate that is on par with that of baseline optimizers. It is worth noting that CO2 can seamlessly integrate with widely adopted ZeRO-series optimizers (Rajbhandari et al., 2020), which are well-established for their capacity to reduce memory consumption within data-parallel distributed training scenarios.

We evaluate CO2 by testing it on various tasks such as image classification, semantic segmentation, point cloud processing, autoregressive language modeling, and bidirectional language modeling. Our thorough assessment demonstrates that CO2 performs equally well in terms of convergence speed and generalization capabilities when compared to other optimization methods.

Our primary contributions can be summarized as follows:

- **Outstanding scalability.** CO2 enables outstanding scalability by fully overlapping communication with computation. This approach excels in large-scale multi-node clusters even with limited communication bandwidth.
- **Good convergence and generalization performance.** We empirically demonstrate that CO2 achieves convergence and generalization performance that is close to well-established baseline optimizers, such as SGD and Adamw. Our results underscore the robustness of CO2 across a diverse range of tasks and datasets.
- **Theoretical convergence analysis.** We provide a rigorous theoretical analysis that guarantees the convergence of CO2. This theoretical foundation enhances our understanding of the practical effectiveness of our approach.
- **Compatibility with ZeRO-series optimizers.** CO2 can integrate with ZeRO-series optimizers to reduce memory usage for large-scale model training.

## 2 RELATED WORK

**Local Updating.** To our known, local updating in distributed deep learning can date back to Zhang et al. (2016). Subsequently, extensive research has delved into local updating methods. Local-SGD (Stich, 2018) provided rigorous theoretical convergence proof for local updating methods. Post-Local-SGD (Lin et al., 2018) and Overlap-Local-SGD (Wang et al., 2020) reduced communication costs but raised concerns about convergence. SlowMo (Wang et al., 2019) improved

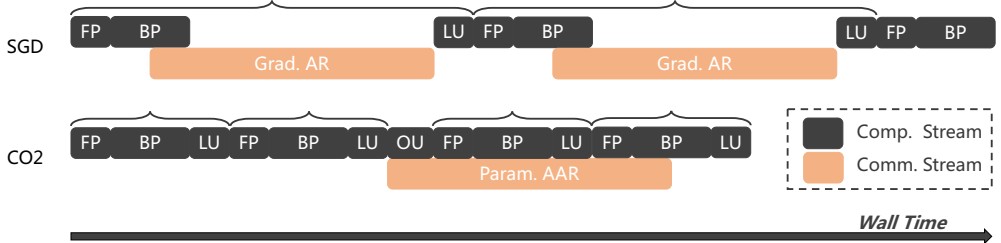

FP: Forward Pass   BP: Backward Pass   LU: Local Update   OU: Outer Update   AAR: Asynchronous All-Reduce

Figure 1: **Visualization of CO2 and SGD**. We exemplify the mechanism of CO2 with a local step count $\tau = 2$. This configuration dictates that the outer update starts after every two local steps, concurrently launching an AAR operation on model parameters. This strategy is made to make the full overlap of AAR communication with local computation possible. CO2 can effectively reduce the wall time required for training compared to the conventional SGD in DDP paradigm.

convergence stability and Cooperative SGD (Wang & Joshi, 2021) presented a unified framework of communication-efficient SGD algorithms. However, communication overhead remains a challenge. Our proposed CO2 introduces asynchronism to overlap parameter synchronization and local computation, achieving 100% scalability with good convergence performance.

**Asynchronism.** There is a longstanding history of asynchronous distributed training techniques (Chen et al., 2016). Recently, Zhang et al. (2015); Barkai et al. (2019) have introduced methods that penalize gradient staleness in asynchronous SGD by employing well-defined gap metrics. Dutta et al. (2021) have delved into the trade-offs between staleness and convergence within the context of asynchronous training. Moreover, work like Cohen et al. (2021); Mishchenko et al. (2022), through rigorous theoretical proofs and empirical experiments, have established that asynchronous SGD can achieve convergence comparable to that of mini-batch SGD, even in the presence of arbitrary gradient delays. In a related vein, Su et al. (2022) proposed GBA as an approach to seamlessly transition between synchronous and asynchronous training, for recommendation models. Koloskova et al. (2022) provided enhanced convergence guarantees for asynchronous SGD in the context of distributed federated learning. These contributions collectively advance our understanding of asynchronous training techniques in distributed settings. Our method, as a comparison, only introduces one-step asynchronous staleness in the outer loop, thus obtaining more robust convergence and performance.

**Efficient Communication.** To improve communication efficiency in synchronizing gradients in distributed training, previous methods often utilized quantization techniques to reduce the communication overhead, such as 1-bit Adam (Tang et al., 2021), 1-bit LAMB (Li et al., 2021), and 0/1 Adam (Lu et al., 2022). In a distinct vein, Assran et al. (2019) proposed stochastic gradient push to perform approximate distributed averaging at each iteration, which offers an alternative dimension for accelerating communication. These methods significantly enhance communication efficiency by mitigating communication overhead within a single iteration, which is orthogonal to our proposed method. More substantial benefits can be realized when integrating with these methods.

## 3 METHOD

In this section, we start with an overview of our proposed CO2 and share two novel techniques that enhance its convergence and training stability. Additionally, we provide a mathematical proof of convergence, which proves that CO2 is capable of achieving a convergence rate that is comparable to those of previously widely used optimizers.

### 3.1 THE OVERVIEW

For the distributed data-parallel training of a large-scale deep neural network, the target can be formulated as:

$$\min_x \frac{1}{G} \sum_{i=1}^{G} \mathbb{E}_{\zeta_i \sim D_i} L^{(i)}(x; \zeta_i), \tag{1}$$

where the objective is to minimize a function with respect to the parameters $x \in \mathbb{R}^n$. The optimization problem involves a summation over $G$ workers, each indexed by $i$. For each worker $i$, there are loss function $L^{(i)}$ and data samples $\zeta_i$ drawn from the distribution $D_i$.

In the routine DDP training, model parameters are replicated across all workers within the communication group. Each worker $i$ is then provided with distinct batches of data samples and independently performs forward and backward passes to compute distinct losses $L^{(i)}$ and gradients $g^{(i)}$. To ensure uniformity of model parameters across all workers within the same DDP communication group, the gradients on all workers are exactly synchronized through an *all-reduce* operation, resulting in the aggregated gradient $g$, i.e., $g = \frac{1}{G}\sum_{i=1}^{G} g^{(i)}$. This aggregated gradient is then utilized to update the model parameters across all workers, ensuring consistency of parameter values across each worker.

Local-updating methods, such as *Local-SGD* or other variants, structure the overall training process into two discernible phases: the inner loop and the outer loop. The inner loop unfolds within each worker and encompasses localized updates. Within this phase, every worker maintains its dedicated set of model parameters, facilitating autonomous updates without the need for gradient synchronization at every step. Upon the completion of $\tau$ local updates, the training process transitions to the outer loop. In this phase, an *all-reduce* operation on model parameters is executed to exactly synchronize the model parameters across all computational workers, i.e., $x = \frac{1}{G}\sum_{i=1}^{G} x_{t,\tau}^{(i)}$, ensuring convergence alignment. Recent work (Wang et al., 2019) introduces additional outer iterates to enhance the convergence of communication-efficient base optimizers, which has been proven effective.

CO2 introduces asynchronous *all-reduce* on model parameters within the outer loop to achieve full overlap between communication and computation throughout the entire training procedure. As delineated in Algorithm 1, during the outer iteration $t$, all workers independently perform local updates involving the FP, BP, and LU on their respective worker nodes. The inner loop iterates $\tau$ steps, from $k = 0$ to $k = \tau - 1$. Notably, during this process, no gradient synchronization takes place among the workers, resulting in distinct model parameters $x_{t,\tau}^{(i)}$ on each worker $i$ at the conclusion of the inner loop. This necessitates a synchronization operation to harmonize the model parameters across all workers.

---

**Algorithm 1:** CO2 Algorithm

---

1 **Input:** Data samples $\zeta^{(i)}$ on worker $i$; Inner learning rate $\gamma_t$; Inner loop steps $\tau$; Outer learning rate $\alpha$; Outer momentum factor $\beta$; Outer loop steps $T$; Initial outer momentum $m_0 = 0$.

2 **for** $t \in \{0, 1, \cdots, T-1\}$ **do**

3     **for** $k \in \{0, 1, \cdots, \tau-1\}$ *on worker $i$* **do**

4        FP & BP: $g_{t,k}^{(i)} = \nabla L^{(i)}(x_{t,k}^{(i)}; \zeta_{t,k}^{(i)})$

5        LU: $x_{t,k+1}^{(i)} = x_{t,k}^{(i)} - \gamma_t g_{t,k}^{(i)}$

6     **end**

7     Launch AAR for $x_{t,\tau}^{(i)}$: $AAR(x_{t,\tau}^{(i)})$

8     **if** *is_completed*$(AAR(x_{t-1,\tau}^{(i)}))$ *is not True* **then**

9        $x_{t-1,\tau} = \text{wait}(AAR(x_{t-1,\tau}^{(i)}))$

10     **end**

11     Update staleness gap:
$\Lambda_t = \frac{\|x_{t,0} - x_{t-1,0}\|}{\tau\|x_{t-1,1} - x_{t-1,0}\|} + \mathbf{1}^n$

12     Update one-step stale outer momentum:
$m_t = \beta m_{t-1} + \frac{1}{\Lambda_t} \cdot (x_{t-1,0} - x_{t-1,\tau})$

13     Update outer iterates:
$x_{t+1,0} = x_{t,0} - \alpha \cdot \text{Clip}(m_t, \phi)$

14 **end**

---

Consequently, upon the completion of inner loop, CO2 launches an *all-reduce* operation on $x_{t,\tau}^{(i)}$ for each worker $i$ to synchronize them, *i.e.,* $\text{AAR}(x_{t,\tau}^{(i)})$. In contrast to conventional methods, this *all-reduce* is asynchronous with subsequent computations, which means the subsequent computations are not dependent upon the results obtained by $\text{AAR}(x_{t,\tau}^{(i)})$. Instead, it leverages the outdated model parameters $x_{t-1,\tau}^{(i)}$, which are subjected to a previous AAR operation in the preceding outer iteration $t-1$. It is worth noting that, at this moment, $\text{AAR}(x_{t-1,\tau}^{(i)})$ may not have finished. Therefore, a waiting operation wait$(\cdot)$ should be conducted if it is checked not completed, this guarantees the execution of $\text{AAR}(x_{t-1,\tau}^{(i)})$. The resulting $x_{t-1,\tau}$ is subsequently employed for updating the stale outer momentum, which is then applied to perform an outer iterate to obtain the initial parameters $x_{t+1,0}$ for next outer loop. For the special case of the first outer iteration, lines 8-13 in Algorithm 1 are skipped, since there is no outdated parameters to be used for updating outer iterates. For each respective outer loop, the stale outer momentum lags by only one step compared to using the new version of $x_{t,\tau}$. This strategy introduces minimal one-step staleness in the outer loop and has a subtle impact on network convergence.

The AAR operation runs in parallel with the subsequent outer updates as well as local computations in the next outer loop. Given the practical considerations of cluster sizes and variable communication conditions, configuring an appropriate number of local steps $\tau$ enables the seamless overlap of the entire asynchronous communication with computations.

## 3.2 STALENESS GAP PENALTY

CO2 achieves full overlap between communication and computation through one-step asynchronous communication, enhancing scalability up to 100% with an appropriate number of local steps. However, the asynchronous nature adds noise to the optimization process, potentially causing discrepancies in training loss and generalization accuracy. To compensate, we propose a staleness gap penalty mechanism, which requires an accurate metric to quantify the staleness gap of outer momentum.

Recall the formulation in (1), where $f_i(x) = \mathbb{E}_{\zeta_i \sim D_i} L^{(i)}(x; \zeta_i)$ is the local objective function at worker $i$. Before constructing the staleness gap, we first assume that each $f_i(x)$ is $L$-Lipschitz as stated in Assumption 3.1.

**Assumption 3.1** *For some existed $L > 0$, there is $\|\nabla f_i(x) - \nabla f_i(y)\| \leq L\|x - y\|$, for all inputs $x, y \in \mathbb{R}^n$ and $i \in \{1, 2, \cdots, G\}$.*

The Lipschitz assumption stated above is intuitive, as it suggests that a viable alternative metric for precisely quantifying gradient differences is to consider the distinctions in model parameters. This insight serves as motivation for us to employ the model parameters from different versions as indicators of the staleness gap for outer momentum.

To quantify the staleness gap in outer momentum, a straightforward approach involves computing the difference in outer momentum between different iterations, such as iterations $t + 1$ and $t$, and normalizing this difference to assess its magnitude. Considering $m_{t+1} = \beta m_t + (x_{t,0} - x_{t,\tau})$ and $m_t = \beta m_{t-1} + (x_{t-1,0} - x_{t-1,\tau})$, the staleness gap of $m_t$ in comparison to $m_{t+1}$ can be straightforwardly represented as:

$$\|m_{t+1} - m_t\| = \|\beta(m_t - m_{t-1}) + (x_{t,0} - x_{t,\tau}) - (x_{t-1,0} - x_{t-1,\tau})\|$$
$$\leq \beta\|(m_t - m_{t-1})\| + \|x_{t,0} - x_{t-1,0}\| + \|x_{t,\tau} - x_{t-1,\tau}\|. \quad (2)$$

It is essential to emphasize that at the initiation of iteration $t$, the ongoing asynchronous *all-reduce* operation involving $x_{t,\tau}$ is generally in progress and has may not reached completion. This situation implies that a globally averaged value for $x_{t,\tau}$, accessible for all workers, is unavailable. Considering the outcomes presented in (2), a suitable and readily accessible metric for measuring the staleness gap, denoted as $\Lambda$, can be defined as:

**Definition 3.1** *The staleness gap of the outer momentum $\Lambda_t \in \mathbb{R}^n$ at step $t$ is quantified as the ratio of the model displacement within the outer loop to the maximum distance the parameters can traverse within the inner loop, in iteration $t - 1$. This can be formally articulated as follows:*

$$\Lambda_t = \frac{\|x_{t,0} - x_{t-1,0}\|}{\tau\|x_{t-1,1} - x_{t-1,0}\|} + \mathbf{1}^n. \quad (3)$$

In (3), the term $\|x_{t,0} - x_{t-1,0}\|$ denotes the magnitude of the model parameter changes within the outer loop at iteration $t - 1$, which occurred with the presence of stale outer momentum. $\tau\|x_{t-1,1} - x_{t-1,0}\|$ represents the maximum distance that the model parameters can traverse in inner steps without utilizing stale outer momentum. Typically, during the training process, both the gradient and the learning rate within the inner loop experience a decay, leading to a reduction in the model parameter distance within a single inner step. This behavior elucidates why $\|x_{t-1,1} - x_{t-1,0}\|$ signifies the maximum distance achievable in a single inner step. The notation $\mathbf{1}^n$ signifies the computation of $\Lambda$ is parameter-wise, where $n$ corresponds to the number of parameters. All computations within (3) are conducted element-wise. At each iteration, we recalculate the staleness gap by performing (3), and then penalize it when updating the outer momentum via: $m_t = \beta m_{t-1} + \frac{1}{\Lambda_t} \cdot (x_{t-1,0} - x_{t-1,\tau})$.

## 3.3 OUTER MOMENTUM CLIPPING

Ensuring training stability is of paramount importance, particularly for large language models. We empirically find that the asynchronism introduced by CO2 can occasionally have adverse effects on

training stability. To mitigate the potential issues related to unusual values in outer momentum and enhance overall training stability, we have implemented a coordinate-wise outer momentum clipping as $\text{Clip}(\chi, \phi) = \max\{-\phi, \min\{\chi, \phi\}\}$, which is used to update the outer iterates as $\boldsymbol{x}_{t+1,0} = \boldsymbol{x}_{t,0} - \alpha \cdot \text{Clip}(\boldsymbol{m}_t, \phi)$, where $\phi$ denotes the clipping threshold, which satisfies $\phi > 0$.

### 3.4 CONVERGENCE ANALYSIS

Considering the staleness introduced by our approach, it is imperative to establish its convergence guarantee from a theoretical standpoint. To do so, we begin by presenting the following assumptions, which align with standard practices in this domain.

**Assumption 3.2** *For all $i \in \{1, 2, \ldots, G\}$, there exists a finite positive constant $\sigma^2$ such that $\mathbb{E}_{\zeta \sim D_i} \left\| \nabla L^{(i)}(\boldsymbol{x}; \zeta) - \nabla f_i(\boldsymbol{x}) \right\|^2 \leq \sigma^2$, i.e., the variance of $f_i(\boldsymbol{x})$ is bounded.*

**Assumption 3.3** *There exists a finite positive constant $V$ such that $\mathbb{E} \left\| \boldsymbol{g}_{t,k} - \mathbb{E}[\boldsymbol{g}_{t,k}] \right\|^2 \leq V$.*

Under these assumptions, our convergence results are given as below, where the detailed proof can be found in Appendix A.4.

**Theorem 1** *If we take $\lambda = 1/\Lambda_t, \gamma_t = \gamma$ and ignore the clip operation, such that $\bar{\lambda} = \alpha\lambda, \frac{\bar{\lambda}\gamma}{1-\beta} = \sqrt{\frac{G}{T\tau}}$ and $T\tau \geq GL^2 \left(1 + \sqrt{3} \max\left\{ \frac{3\tau(1-\beta-\alpha)}{\alpha}, \frac{4\tau\beta}{1-\beta}, 1 \right\}\right)$, then under Assumptions 3.1, 3.2, 3.3 and $\frac{1}{G}\sum_{i=1}^{G} \|\nabla f(\mathbf{x}) - \nabla f_i(\mathbf{x})\|^2 \leq \delta^2$, where $\delta$ is a positive finite constant, we have:*

$$\frac{1}{T\tau}\sum_{t=0}^{T-1}\sum_{k=0}^{\tau-1} \mathbb{E}\left\| \nabla f\left(\boldsymbol{x}_{t,k}\right) \right\|^2 = \mathcal{O}\left(\frac{1}{\sqrt{GT\tau}}\right) + \mathcal{O}\left(\frac{G\tau}{T}\right), \quad (4)$$

where $f = \frac{1}{G}\sum_{i=1}^{G} f_i(x)$. The theorem indicates that, when the total steps $T\tau$ is sufficiently large, i.e, $T \gg G^3\tau^3$, the RHS is dominated by $\mathcal{O}\left(\frac{1}{\sqrt{GT\tau}}\right)$. So, when the number of workers is $G$ times more, we only need $G$ times less total steps to achieve the same error.

## 4 EXPERIMENTS

We conducted a thorough assessment of CO2 in a range of deep learning applications spanning computer vision (CV) and natural language processing (NLP). Our testing encompassed an array of model architectures, including convolution networks, transformer networks, and linear transformer networks. For more specific information regarding the tasks, models, their categorizations, parameter counts, and datasets used, please refer to Table 5 in Appendix A. It is worth noting that the TransNormer-LLM (7B) (Qin et al., 2023b; 2024a;b) experiment mainly aims to evaluate CO2's scalability, so we only pre-train it on a relatively small dataset WikiText-103 to assess the convergence.

### 4.1 EXPERIMENTAL SETUP

We compared CO2 against other state-of-the-art optimization techniques, including SGD, Adamw, Local-SGD/Adamw, Overlap-Local-SGD/Adamw, and SlowMo. Hyperparameters for each approach were meticulously tuned for maximum performance. For CO2, we conducted an extensive hyperparameter search for $\tau$, ranging from $\{1, 3, 6, 12, 24, 48, 96, 192\}$ to find the optimal balance between efficiency and performance. All experiments were executed five times with distinct random seeds to ensure robust results. For hardware, software, and hyperparameter details, see Appendix A.1 and A.2.

**Image Classification (IC).** We train ResNet-50 (He et al., 2016; Zhou et al., 2020), ViT (Dosovitskiy et al., 2020; Tang et al., 2024), and VVT (Sun et al., 2023) on ImageNet-1K dataset, using eight A100 nodes with 64 GPUs. For the training of ResNet-50, we use a total mini-batch size of 8192 and train for 90 epochs with a cosine learning rate schedule. For the training of ViT and VVT, they closely adhere to analogous hyperparameters. We use an uniform total batch size of 2048 for both and train for 300 epochs.

**Semantic Segmentation (SS).** We leverage the pre-trained VVT on ImageNet-1K as the backbone model, adopt semantic feature pyramid network (Semantic FPN) (Lin et al., 2016) as the decoder and finetune on the ADE20K dataset, using four 3090 nodes with 32 GPUs.

Table 1: **Convergence performance on CV tasks.** CO2 performs better than other local-updating methods with a clear margin and is comparable to standard optimizers such as SGD/Adamw.

| Task | Model | SGD (Adamw) | Local-SGD(Adamw) | Overlap-Local-SGD(Adamw) | SlowMo | CO2 |
|---|---|---|---|---|---|---|
| IC | ResNet-50 | 76.92 ($\pm$ 0.05) | 75.57 ($\pm$ 0.76) | 76.28 ($\pm$ 0.18) | 77.12 ($\pm$ 0.11) | 77.14 ($\pm$ 0.09) |
| | ViT (Base) | 81.33 ($\pm$ 0.04) | 78.43 ($\pm$ 0.22) | 78.04 ($\pm$ 0.35) | 79.83 ($\pm$ 0.16) | 80.95 ($\pm$ 0.08) |
| | VVT (Large) | 83.64 ($\pm$ 0.06) | 81.09 ($\pm$ 1.15) | 80.33 ($\pm$ 0.49) | 82.75 ($\pm$ 0.27) | 83.38 ($\pm$ 0.06) |
| SS | VVT (Large) | 47.82 ($\pm$ 0.05) | 44.25 ($\pm$ 2.24) | 45.21 ($\pm$ 1.36) | 47.51 ($\pm$ 0.12) | 47.80 ($\pm$ 0.11) |
| PC | Point-MAE | 68.56 ($\pm$ 0.08) | 64.25 ($\pm$ 2.11) | 63.78 ($\pm$ 1.92) | 68.69 ($\pm$ 0.32) | 68.89 ($\pm$ 0.39) |

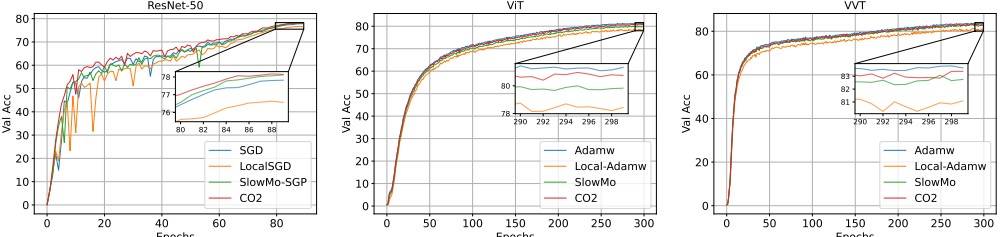

Figure 2: **Validation curves for image classification tasks.** Three models ResNet-50, ViT, and VVT are trained on ImageNet-1K for 90, 300 and 300 epochs, respectively. Our CO2 exhibits robust convergence and good generalization performance when compared to other existing methods.

**3D Point Cloud (PC) Reconstruction.** We pre-train the Point-MAE (Pang et al., 2022; Li et al., 2023) via reconstruction task using ShapeNet dataset (Chang et al., 2015). We use a point number of 1024, a total batch size of 256, epoch number of 300, train on four 3090 nodes with 32 GPUs.

**Autoregressive Language Modeling (ALM).** For ALM tasks, we train autoregressive GPT-2 models with 125M (Small), 355M (Medium), and 770M (Large) parameters on the OpenWebText dataset (Radford et al., 2019). All experiments use a context length of 1024 and a total batch size of 480, training on eight A100 nodes with 64 GPUs. Additionally, we train the large linear transformer (Qin et al., 2022a;b; 2023a;c) model TransNormer-LLM (7B) on WikiText-103 for 100K iterations on sixteen A100 nodes with 128 GPUs, in order to verify the performance of CO2 on large language models. The ZeRO-2 optimizer is integrated with CO2 to lessen memory redundancy during training TransNormer-LLM (7B).

**Bidirectional Language Modeling (BLM).** For BLM tasks, we train a RoBERTa (Large) (Liu et al., 2019) model on a combined dataset consisting of Wikipedia and BookCorpus (Wettig et al., 2022) for approximately 3 epochs, using a total batch size of 4096 and sequence length of 128, train on four 3090 nodes with 32 GPUs.

## 4.2 CONVERGENCE AND GENERALIZATION

**Computer vision results.** Table 1 shows the results of convergence and generalization in computer vision tasks. CO2 optimizer consistently outperforms other efficient optimization techniques in terms of Top-1 validation accuracy for ResNet-50, ViT, and VVT architectures. Although CO2 has a slightly lower accuracy than the Adamw for both ViT and VVT, the reduction is marginal and falls within the expected range of random fluctuations. In addition to its good performance, CO2 achieves more efficient communication, including the potential for zero communication delay compared to the baseline methods. A similar trend is observed for VVT in semantic segmentation tasks. CO2 demonstrates a notable performance advantage over (Overlap-)Local-SGD and SlowMo, with only a slight performance gap of 0.02% when compared to Adamw. For 3D point cloud reconstruction, our proposed CO2 outperforms all other counterparts, exceeds the baseline Adamw by 0.33%.

We also present validation curves that track training progress of these models on IC tasks. As shown in Fig. 2, CO2 features a more stable validation curve for ResNet-50 and maintains consistently higher accuracy curves compared to (Overlap-)Local-Adamw and SlowMo for ViT and VVT.

**Natural language processing results.** Table 2 and Fig. 3 presents the convergence and generalization outcomes for natural language processing tasks. The table shows that CO2 consistently outperforms alternative methods across GPT-2 variants (Small, Medium, and Large) for ALM tasks, resulting in lower validation perplexity values as the model size scales. Furthermore, CO2 achieves the best

Table 2: **Convergence performance on NLP tasks.** Quantitative perplexity (PPL) results for GPT-2, TransNormer-LLM and RoBERTa are presented. CO2 shows lower perplexity (lower is better) than the baseline Adamw in all these experiments.

| Task | Model | Adamw | Local-Adamw | Overlap-Local-Adamw | SlowMo | CO2 |
|------|-------|-------|-------------|---------------------|--------|-----|
| ALM | GPT-2 (Small) | 7.44 ($\pm$ 0.36) | 7.95 ($\pm$ 2.04) | 8.11 ($\pm$ 1.03) | 7.34 ($\pm$ 0.89) | 7.37 ($\pm$ 0.73) |
| | GPT-2 (Medium) | 6.61 ($\pm$ 0.53) | 7.49 ($\pm$ 1.87) | 7.26 ($\pm$ 1.44) | 6.41 ($\pm$ 0.65) | 6.36 ($\pm$ 0.66) |
| | GPT-2 (Large) | 6.26 ($\pm$ 0.58) | 7.00 ($\pm$ 1.91) | 7.18 ($\pm$ 0.98) | 6.29 ($\pm$ 0.61) | 6.13 ($\pm$ 0.52) |
| | TN-LLM (7B) | 16.82 ($\pm$ 0.86) | 18.63 ($\pm$ 3.13) | 17.83 ($\pm$ 2.95) | 16.95 ($\pm$ 1.01) | 16.78 ($\pm$ 0.95) |
| BLM | RoBERTa (Large) | 3.96 ($\pm$ 0.37) | 4.38 ($\pm$ 0.83) | 4.52 ($\pm$ 1.42) | 3.98 ($\pm$ 0.85) | 3.95 ($\pm$ 0.96) |

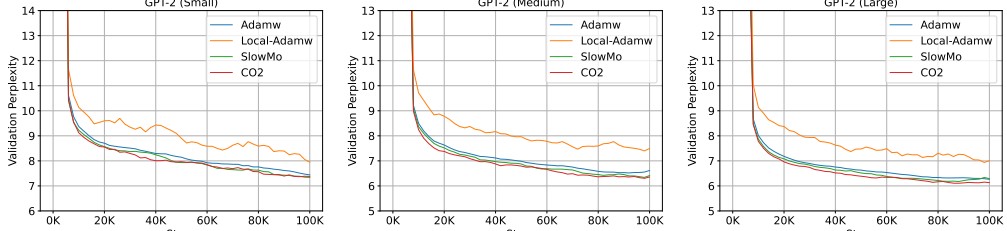

Figure 3: **Validation curves for autoregressive language tasks.** We train GPT-2 on OpenWebText for 100K steps in three sizes: 125M (Small), 355M (Medium), and 770M (Large). CO2 exhibits robust convergence and the best generalization performance when compared to other existing methods.

training perplexity on the large TransNormer-LLM (7B) model, when trained on a relatively small dataset. We also give validation curves with respect to relative time (in seconds) for GPT-2 models of varying sizes as illustrated in Fig. 7, Appendix A.3.

For BLM tasks, CO2 demonstrates comparable performance to the standard Adamw and SlowMo algorithms, surpassing (Overlap-)Local-Adamw by a significant margin. These results emphasize the capacity of CO2 to optimize encoder-only transformer models without compromising accuracy.

## 4.3 SCALABILITY

We tested our proposed CO2 using up to 16 DGX-A100 servers to train TransNormer-LLM with 7B parameters. Each server has 8 A100 GPUs interconnected via NVSwitch for an inter-GPU bandwidth of 600GBps. We evaluated CO2's scalability under different cluster communication conditions using two kinds of inter-node network configurations. The first uses RoCE solution with 8 RDMA adapters per server for inter-server communication at 800Gbps. The second uses TCP/IP Ethernet with significantly lower bandwidth at around 80Gbps. Note that, to mitigate fluctuations, the presented throughput results are averaged values extracted from iterations 100 to 200 for each experiment.

We evaluated two methods, Adamw and CO2, on RoCE RDMA and TCP/IP networks. On the RoCE RDMA network, CO2's communication advantage became prominent in larger clusters (more than 64 GPUs) resulting in higher throughput than Adamw. On the TCP/IP network, CO2 performed similarly to its performance on the RoCE RDMA network when $\tau$ was set to 48. However, Adamw performed poorly due to pronounced communication latency. We also test the influence of $\tau$ on communication efficiency, with varying levels of overlap. Results are shown in Fig. 4(a).

Using CO2 with TCP/IP and $\tau = 12$ caused a moderate reduction in speed and scalability, but it still outperformed Adamw on TCP/IP due to overlap between communication and computation. Transitioning from 8 GPUs to 16 GPUs caused a notable drop in throughput values for both CO2 and Adamw due to the diminished inter-node connectivity.

Table 3 shows throughput and scalability of CO2 and Adamw. Scalability ratios were calculated for different communication scenarios, specifically when transitioning from 16 GPUs to 128 GPUs. CO2 exhibits scalability ratios exceeding 1, mainly due to inherent measurement fluctuations.

## 4.4 ABLATION STUDY

We conducted ablation studies on CO2 using a small-scale GPT-2 (Small) model across four servers, each with eight 3090 GPUs. The experiments lasted for 100K steps.

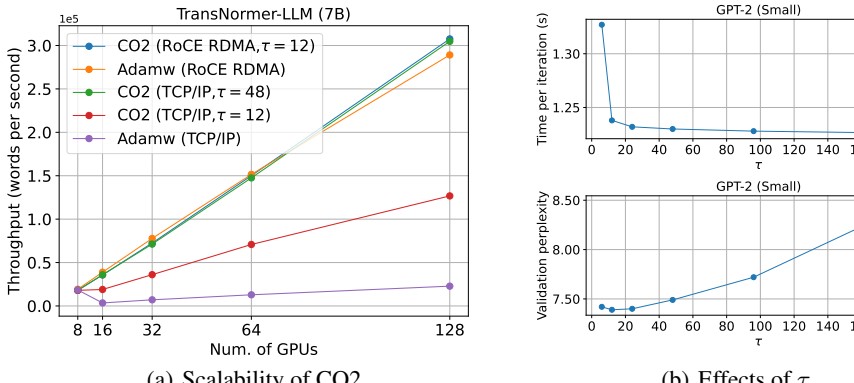

(a) Scalability of CO2.                    (b) Effects of $\tau$.

Figure 4: (a): **Scalability of CO2.** Throughput (words/sec) results on distinctive inter-node network configurations are presented. CO2 exhibits pecfect $100\%$ scalability on both configurations. (b): **Effects of $\tau$.** Training speed and generalization performance results w.r.t. $\tau$ are presented. A larger value of $\tau$ leads higher communication efficiency but worse generalization behaviors.

Table 3: **Quantitative scalability performance of CO2.** Throughput (words/sec) results of CO2 and Adamw from 1 to 16 DGX-A100 servers are presented. Scalability ratio records the scalability from 2 to 16 nodes to take into account the inter-node connections. With suitable configuration of $\tau$, CO2 can reach the scalability ratio of 1, which outperforms the baseline Adamw.

| Ethernet | Method | Throughput | | | | | Scalability Ratio $(16 \rightarrow 128)$ |
|---|---|---|---|---|---|---|---|
| | | 8 GPUs | 16 GPUs | 32 GPUs | 64 GPUs | 128 GPUs | |
| RDMA | CO2 ($\tau$=12) | 17980 ($\pm$ 48) | 35692 ($\pm$ 126) | 72491 ($\pm$ 183) | 150073 ($\pm$ 362) | 307557 ($\pm$ 617) | 1.08 |
| | Adamw | 19276 ($\pm$ 59) | 38888 ($\pm$ 118) | 77782 ($\pm$ 93) | 151554 ($\pm$ 209) | 289106 ($\pm$ 423) | 0.93 |
| TCP/IP | CO2 ($\tau$=48) | 18090 ($\pm$ 95) | 35969 ($\pm$ 193) | 71249 ($\pm$ 179) | 147507 ($\pm$ 315) | 304736 ($\pm$ 729) | 1.06 |
| | CO2 ($\tau$=12) | 17995 ($\pm$ 72) | 18975 ($\pm$ 108) | 36095 ($\pm$ 151) | 70839 ($\pm$ 373) | 129865 ($\pm$ 564) | 0.86 |
| | Adamw | 18444 ($\pm$ 49) | 3488 ($\pm$ 115) | 7077 ($\pm$ 127) | 12855 ($\pm$ 308) | 22810 ($\pm$ 526) | 0.82 |

**Performance and scalability w.r.t. $\tau$.** We conducted experiments to examine time per iteration and validation perplexity across different values of $\tau$. Increasing $\tau$ generally leads to faster training but higher perplexity. However, at $\tau = 12$, computation and communication overlap, resulting in a significant acceleration. Larger $\tau$ values benefit from reduced communication frequency. $\tau = 12$ also yields a slightly lower perplexity, which can be attributed to the regularization effect.

**Staleness gap penalty and outer momentum clipping.** Table 4 shows that incorporating the staleness gap penalty has a more significant positive impact than outer momentum clipping. The latter mainly targets training stability rather than performance improvement.

Table 4: **Ablation results on staleness gap penalty and outer momentum clipping.** We pre-train GPT-2 (Small) with 100K steps for ablation. Both train and validation perplexity results indicate that staleness gap penalty has a noteworthy improvement for convergence performance.

| Model | Steps | Metric | CO2 | CO2 w/o Penalty | CO2 w/o Clipping |
|---|---|---|---|---|---|
| GPT-2 (Small) | 100K | Train PPL | 7.36 | 7.52 | 7.39 |
| | | Validation PPL | 7.39 | 7.56 | 7.42 |

## 5 CONCLUSION

We proposed CO2, a method for large-scale distributed training on clusters with low-cost and limited-speed interconnections. CO2 enables exceptional scalability through local-updating and one-step asynchronous communication. Mathematical analysis is provided on its convergence rate. Our experiments in computer vision and natural language processing demonstrate CO2's ability to achieve $100\%$ scalability even on clusters with very limited communication bandwidth. We have enhanced CO2's convergence and stability through the introduction of two techniques: the staleness gap penalty and outer momentum clipping. CO2 is highly compatible with established optimizers and can be integrated with ZeRO-series optimizers to reduce memory consumption for large model training.

ACKNOWLEDGEMENTS

This work is partially supported by the National Key R&D Program of China (NO.2022ZD0160100).

ETHICS STATEMENT

Given the widespread adoption of large language models trained on extensive corpora, the significance of distributed training across expansive clusters has grown substantially. Inefficient training practices result in substantial computational resource wastage, leading to elevated economic costs and an increased emission of $CO_2$, a significant contributor to the global greenhouse effect. In the context of distributed training for neural networks on GPU clusters, efficient communication plays a pivotal role. However, existing methods still grapple with low communication efficiency, particularly on modern, large-scale GPU clusters characterized by limited inter-node connections. Our proposed approach addresses this challenge by advocating for the full overlap of communication and computation, thereby substantially expediting model training even in scenarios with severely restricted communication capabilities. This not only reduces training time but also translates into cost savings and environmental benefits, as it contributes to mitigating the impact on the environment.

In addition, our proposed method is relatively general and not limited to specific models or tasks. It can be well extended to other tasks not included in this paper, such as stable diffusion for image generation, object detection for autonomous driving, etc.

REPRODUCIBILITY STATEMENT

For the sake of facilitating reproducibility in our experiments, we provide a comprehensive listing of the hardware and software components employed in our study. This information is thoughtfully presented in Appendix A and encompasses details such as the specifications of GPUs, inter-node communication configurations, and the versions of critical software components including PyTorch, CUDA, cuDNN, and NCCL. Additionally, we give links to the open-source code repositories that were instrumental in the execution of experiments across a spectrum of tasks in the paper. These concerted efforts reinforce the transparency and replicability of our research.

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

# Appendix

## A  EXPERIMENT DETAILS

We show specific experiment plans regarding the tasks, models, their categorizations, parameter counts, and datasets used in Table .5.

Table 5: **Tasks, Models, and Datasets implemented in the experiments.** IC: image classification, SS: semantic segmentation, PC: Point Cloud, ALM: autoregressive language modeling, BLM: bidirectional language model. TN-LLM: TransNormer-LLM.

| Field | Task | Model | Parameters | Model Type | Dataset |
|-------|------|-------|-----------|-----------|---------|
| CV | IC | ResNet-50 | 25.6M | ConvNet | ImageNet-1k |
| | | ViT (Base) | 86.6M | Transformer | |
| | | VVT (Large) | 61.8M | Linear Transformer | |
| | SS | VVT (Large) | 65.5M | Linear Transformer | ADE20K |
| | PC | Point-MAE | 10.2M | Transformer | ShapeNet |
| NLP | ALM | GPT-2 (Small) | 125M | Transformer (Decoder) | OpenWebText |
| | | GPT-2 (Medium) | 355M | Transformer (Decoder) | |
| | | GPT-2 (Large) | 770M | Transformer (Decoder) | |
| | | TN-LLM (7B) | 6.7B | Linear Transformer | WikiText-103 |
| | BLM | RoBERTa (Large) | 355M | Transformer (Encoder) | Wikipedia+BookCorpus |

### A.1  HARDWARE AND SOFTWARE

**Hardware.**  Our experimental setup comprises up to 16 DGX-A100 servers, with each server featuring 8 A100 GPUs. These GPUs are interconnected via NVSwitch, providing an inter-GPU bandwidth of 600GBps. Two distinct inter-node network configurations are used. The first configuration leverages RoCE (RDMA over Converged Ethernet) solution, employing 8 RoCE RDMA adapters in each server to facilitate inter-server communication with a bandwidth capacity of 800Gbps. The second configuration employs the TCP/IP protocol which operates at a substantially reduced bandwidth compared to RoCE. Specifically, it offers approximately 1/10th the bandwidth, totaling around 80Gbps. Part of experiments are conducted on a 3090 GPU cluser with total 10 servers. Each server is equipmented by eight 3090 GPUs.

**Software.**  Experiments are implemented in PyTorch 1.13.0 with CUDA 11.7, cuDNN 8.0, and NCCL 2.14.3. Our algorithm is developed upon FairScale 0.4.13. The image classification experiments build on https://github.com/huggingface/pytorch-image-models and https://github.com/OpenNLPLab/Vicinity-Vision-Transformer. The semantic segmentation experiments build on https://github.com/whai362/PVT. The 3D point cloud reconstruction experiments build on https://github.com/Pang-Yatian/Point-MAE. The autoregressive language modeling experiments build on https://github.com/karpathy/nanoGPT and https://github.com/facebookresearch/metaseq. The bidirectional language modeling experiments build on https://github.com/princeton-nlp/DinkyTrain.

### A.2  HYPERPARAMETER DETAILS

We detail hyperparameter settings for all tasks in the following:

**Image Classification.**  The image classification experiments are conducted on ImageNet1K dataset, which contains 1.28M training images and 50K validation images from 1, 000 categories. We adopt ResNet-50 (He et al., 2016; Zhou et al., 2020), ViT (Dosovitskiy et al., 2020), and VVT (Sun et al., 2023), which respectively correspond to convolutional network, transformer network, and linear transformer network. For the training of ResNet-50, the total mini-batch size is 8192, and the training runs 90 epochs. We sourced hyperparameters from DeepLearningExamples [1]. The momentum value

---
[1]https://github.com/NVIDIA/DeepLearningExamples

is set to 0.875 and the learning rate is initialized to 0.256 for a batch size of 256, the value is linearly scaled for batch sizes distinct from this reference. The learning rate schedule adheres to a cosine schedule, while a linear warm-up mechanism for the learning rate is introduced for the initial 5 epochs. We set the weight decay to 1/32768 and we do not apply weight decay on batch normalization trainable parameters. A label smoothing coefficient of 0.1 is adopted to enhance model robustness.

For the training of ViT and VVT, they closely adhere to analogous hyperparameters. A uniform total batch size of 2048 is employed, sustained across 300 epochs. A cosine learning rate schedule supplemented by a linear warm-up during the initial 5 epochs is incorporated. A label smoothing coefficient of 0.1 is applied, while weight decay is set at 0.05. We use an initial learning rate of $5 \times 10^{-4}$, this value diminishes with a cosine schedule, with 5 epochs dedicated to the warm-up phase. The augmentation follows previous literature, which includes practices such as random cropping and random horizontal flipping. Throughout the training, all models undergo 300 epochs of training on the training set, adopting a crop size of $224 \times 224$. The evaluation metric is the top-1 accuracy.

**Semantic Segmentation.** For the semantic segmentation task on the ADE20K dataset, the pre-trained ViT and VVT models undergo a finetuning process. The ADE20K dataset is composed of 150 distinct classes, distributed across 20210, 2000, and 3352 images for training, validation, and testing, respectively. We leverage the ViT and VVT models as the foundational backbone, already pre-trained on ImageNet1K. we adopt the Semantic Feature Pyramid Network (Semantic FPN) (Lin et al., 2016) as the decoder. Hyperparameters governing the training of both ViT and VVT models are determined with reference to VVT.

**3D Point Cloud Reconstruction.** We trained the Point-MAE (Pang et al., 2022; Li et al., 2023) via the reconstruction task using ShapeNet dataset (Chang et al., 2015). We further adopt the Chamfer distance as reconstruction loss, as well as a point number of 1024, a batch size of 256, epoch number of 300, cosine learning rate schedule, Adamw optimizer (Loshchilov & Hutter, 2018) with a learning rate of 0.0005, and weight decay of 0.05. The distributed training performance is evaluated not only by the training loss but also by the accuracy in the validation dataset of ModelNet40 (Wu et al., 2015). Detailed reconstruction pre-training and validation settings may be found in Pang et al. (2022); Li et al. (2023).

**Autoregressive Language Modeling.** For ALM tasks, we trained autoregressive GPT-2 models (Radford et al., 2019) on OpenWebText dataset (Radford et al., 2019). Three GPT-2 models with 125M (Small), 355M (Medium), and 770M (Large) parameters are tested to prove the scalability of CO2 . All experiments use the context length of 1024 and use a total batch size of 480. In addition, to further validate the scalability and efficiency of CO2 , we conducted experiments a large efficient language model, namely TransNormer-LLM (7B) for 100k steps on small dataset WikiText-103. ZeRO-series optimizers are integrated with CO2 to lessen memory redundancy during training TransNormer-LLM (7B).

**Bidirectional Language Modeling.** For BLM tasks, we trained a RoBERTa (Large) model on a combined dataset with Wikipedia [2] and BookCorpus (Zhu et al., 2015) for approximately 3 epochs. We utilize a total batch size of 4096 and sequence length of 128. A polynomial learning rate decay scheduler was employed, with a peak learning rate of 2e-3 after 6% of total updates as warm-up.

### A.3 ADDITIONAL EXPERIMENTAL RESULTS

### A.3.1 TRAINING CONVERGENCE CURVES

In main paper, we show validation accuracy curves on classification task of three models, *i.e.,* ResNet-50, ViT and VVT, in Fig. 2, we show their respective training loss curves in Fig. 5. Similarly, Fig.3 demonstrates the validation perplexity curves of GPT-2 with different sizes (Small, Medium and Large) on autoregressive language tasks. We show their respective curves of training perplexity in Fig. 6. These curves further validate the effectiveness of the proposed CO2.

---

[2]https://dumps.wikimedia.org

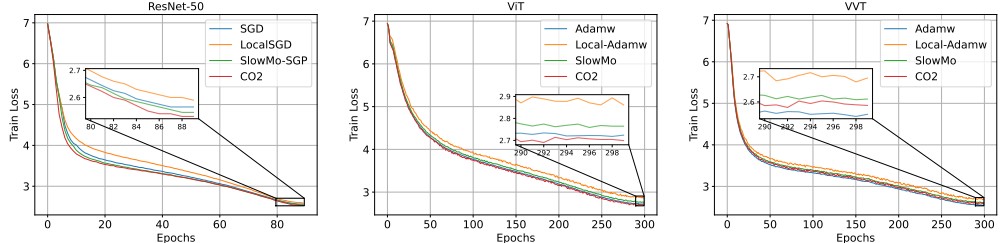

Figure 5: **Training curves for image classification tasks.** Three models ResNet-50, ViT and VVT are trained on ImageNet-1K for 90, 300 and 300 epochs, respectively. Our CO2 exhibits robust convergence compared to other existing methods.

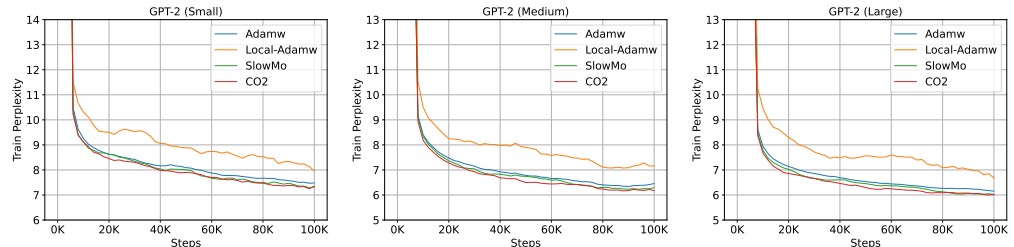

Figure 6: **Training curves for autoregressive language tasks.** Three variants of GPT-2 with 125M (Small), 355M (Medium), and 770M (Large) parameters are pre-trained on OpenWebText for 100K steps. Our CO2 exhibits the fastest convergence rate compared to other existing methods.

### A.3.2 TIME TO ACCURACY CURVES

In order to highlight the benefits of CO2 in accelerating the training process, we give validation curves with respect to relative time (in seconds) for GPT-2 models of varying sizes as illustrated in Fig. 7. The outcomes demonstrate that CO2 excels not only in achieving superior generalization performance but also in minimizing the duration of the training process, thereby affirming its practical utility in distributed training.

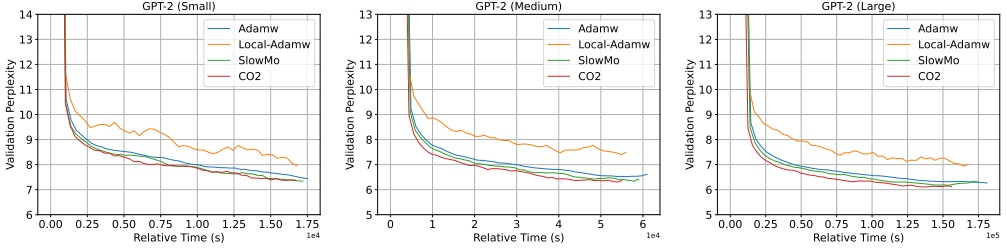

Figure 7: **Validation curves w.r.t. relative time for autoregressive language tasks.** Three variants of GPT-2 with 125M (Small), 355M (Medium), and 770M (Large) parameters are pre-trained on OpenWebText for 100K steps. Our CO2 exhibits the fastest convergence speed on relative time compared to other existing methods.

### A.3.3 CONVERGENCE RESULTS WITH MORE DETAILS

To show advantages of CO2 on the trade-off of training efficiency and convergence performance, we list the corresponding throughput and settings of $\tau$ for both CV and NLP tasks in Table 6 and Table 7. The presented throughput results represent averaged values extracted from iterations 100 to 200 for each experiment. Given the extensive nature of our experiments, convergence outcomes in the paper originate from two distinct clusters. Specifically, the IC and ALM tasks were executed on

Table 6: **Convergence performance on CV tasks.** $CO_2$ performs better than other local methods with a clear margin and is comparable to standard optimizers such as SGD/Adamw. For IC and SS tasks, we present the throughput results in images/sec, the image resolution is $224 \times 224$; For PC task, we present the throughput results in point clouds/sec, each point cloud has 1024 points.

| Task | Model | SGD (Adamw) | Local-SGD(Adamw) | Overlap-Local-SGD(Adamw) | SlowMo | CO2 |
|------|-------|-------------|------------------|--------------------------|--------|-----|
|      |       | Acc / Thpt | Acc / Thpt / $\tau$ | Acc / Thpt / $\tau$ | Acc / Thpt / $\tau$ | Acc / Thpt / $\tau$ |
| IC | ResNet-50 | 76.92 / 108739 | 75.57 / 108758 / 1 | 76.28 / 108765 / 1 | 77.12 / 108741 / 1 | 77.14 / 108753 / 1 |
|    | ViT (Base) | 81.33 / 39422 | 78.43 / 39512 / 3 | 78.04 / 39511 / 3 | 79.83 / 39509 / 3 | 80.95 / 39533 / 3 |
|    | VVT (Large) | 83.64 / 44375 | 81.09 / 44390 / 1 | 80.33 / 44392 / 1 | 82.75 / 44376 / 1 | 83.38 / 44387 / 1 |
| SS | VVT (Large) | 47.82 / 5384 | 44.25 / 5528 / 6 | 45.21 / 5545 / 6 | 47.51 / 5521 / 6 | 47.80 / 5562 / 6 |
| PC | Point-MAE | 68.56 / 5859 | 64.25 / 5931 / 3 | 63.78 / 5950 / 3 | 68.69 / 5904 / 3 | 68.89 / 5956 / 3 |

Table 7: **Convergence performance on NLP tasks.** Quantitative perplexity (PPL) results for GPT-2, TransNormer-LLM and RoBERTa are presented. $CO_2$ shows lower perplexity (lower is better) than the baseline Adamw in all these experiments. For ALM and BLM tasks, we present the throughput results in words/sec. TN-LLM represents TransNormer-LLM.

| Task | Model | Adamw | Local-Adamw | Overlap-Local-Adamw | SlowMo | CO2 |
|------|-------|-------|-------------|---------------------|--------|-----|
|      |       | Acc / Thpt | Acc / Thpt / $\tau$ | Acc / Thpt / $\tau$ | Acc / Thpt / $\tau$ | Acc / Thpt / $\tau$ |
| ALM | GPT-2 (Small) | 7.44 / 6.543e6 | 7.95 / 6.556e6 / 3 | 8.11 / 6.556e6 / 3 | 7.34 / 6.554e6 / 3 | 7.37 / 6.556e6 / 3 |
|     | GPT-2 (Medium) | 6.61 / 2.084e6 | 7.49 / 2.094e6 / 3 | 7.26 / 2.092e6 / 3 | 6.41 / 2.091e6 / 3 | 6.36 / 2.092e6 / 3 |
|     | GPT-2 (Large) | 6.26 / 1.052e6 | 7.00 / 1.059e6 / 6 | 7.18 / 1.057e6 / 6 | 6.29 / 1.053e6 / 6 | 6.13 / 1.056e6 / 6 |
|     | TN-LLM (7B) | 16.82 / 0.281e6 | 18.63 / 0.303e6 / 12 | 17.83 / 0.306e6 / 12 | 16.95 / 0.301e6 / 12 | 16.78 / 0.308e6 / 12 |
| BLM | RoBERTa (Large) | 3.96 / 2262 | 4.38 / 2815 / 6 | 4.52 / 2877 / 6 | 3.98 / 2794 / 6 | 3.95 / 2892 / 6 |

an A100 cluster equipped with RoCE RDMA high-speed inter-node connections, featuring 8 nodes and a total of 64 GPUs. Conversely, the SS, PC, and BLM tasks were conducted on a 3090 cluster with a standard TCP/IP Ethernet inter-node connection, comprising 4 nodes and 32 GPUs. Upon observing tasks trained on the A100 platform, it is evident that the throughput of $CO_2$ surpasses other counterparts, although with a slight advantage. This advantage becomes more pronounced on tasks trained on the 3090 platform. It is reasonable to infer that this advantage will be amplified on larger clusters, especially those with slower inter-node connections.

Prior to commencing large-scale convergence experiments, we meticulously tune the values of $\tau$ for each task using a simple grid search strategy within the range of $\{1, 3, 6, 12, 24, 48, 96, 192\}$ to reach a balance of accuracy and throughput. Hyperparameter tuning experiments are conducted on smaller versions of models when the model to be trained is large, owing to the associated high time and resource costs. We start the tuning of $\tau$ from 1 to larger candidate values given the consideration of high-speed communication on the corresponding platform. It is worth noting that, in addition to $CO_2$, other methods such as Local-SGD/Adamw and SlowMo also require tuning of $\tau$. In practice, we exclusively tune $\tau$ for $CO_2$ and employ the same tuned $\tau$ values for Local-SGD/Adamw and SlowMo. This is because of the similar role played by $\tau$ in these local-updating methods. Employing identical $\tau$ values for different methods on the same task ensures a fair comparison of their convergence and training throughput.

### A.3.4 COMMUNICATION-COMPUTATION OVERLAP RATIO

We conducted quantitative measurements on the time allocated to local computations and asynchronous communication. This assessment specifically targeted $CO_2$ with different $\tau$ configurations on an A100 cluster equipped with 128 GPUs, employing a slower TCP/IP inter-node network. Adhering to the settings outlined in the paper, utilizing TransNormer-LLM (7B), we maintained consistency in our experimental conditions. The measured duration for a single-step local computation was approximately 0.109s, while an all-reduce communication incurred a cost of 1.566s. Subsequently, we computed and calculated the communication/computation overlap ratio for various $\tau$ values, presenting the results in the Table 8.

Table 8: **Communication-computation overlap ratio results.** The varying values of $\tau$ and the corresponding overlap ratios on TransNormer-LLM (7B).

| $\tau$ | 1 | 3 | 6 | 12 | 24 | 48 |
|---|---|---|---|---|---|---|
| **Overlap Ratio** | 6.52% | 20.39% | 41.81% | 83.28% | 100% | 100% |

### A.3.5 SCALABILITY RESULTS ON GPT-3 (13B)

To further effectively demonstrate the scalability advantages of CO2 on large models, we conducted experiments specifically on GPT-3 (13B), a well-known language model with autoregressive transformer architecture. The results are shown in the Table 9, includes throughput comparisons with SlowMo. The results reveal that on GPT-3 (13B), CO2 consistently achieves higher throughput on platforms with different communication conditions.

Table 9: **Throughput results on GPT-3 (13B).** Throughput (words/sec) results of CO2 , SlowMo and Adamw from 1 to 16 DGX-A100 servers are presented. With suitable configuration of $\tau$, CO2 can reach the highest throughput which outperforms SlowMo and the baseline Adamw.

| Ethernet | Method | $\tau$ | Throughput | | | | |
|---|---|---|---|---|---|---|---|
| | | | 8 GPUs | 16 GPUs | 32 GPUs | 64 GPUs | 128 GPUs |
| RDMA | CO2 | 12 | 9468 | 18587 | 37905 | 75682 | 151846 |
| | SlowMo | 12 | 9463 | 18620 | 37467 | 75503 | 148987 |
| | Adamw | \ | 9519 | 19131 | 38824 | 75326 | 147723 |
| TCP/IP | CO2 | 48 | 9508 | 18672 | 38151 | 75573 | 151678 |
| | SlowMo | 48 | 9424 | 18511 | 38092 | 75209 | 146024 |
| | CO2 | 12 | 9324 | 9694 | 18792 | 36968 | 78638 |
| | SlowMo | 12 | 9359 | 9628 | 18085 | 33864 | 66908 |
| | Adamw | \ | 9482 | 2283 | 4192 | 7984 | 13832 |

### A.3.6 PENALTY PERFORMANCE ON NON-DECAY LEARNING RATE

To evaluate the performance of staleness gap penalty on non-decay learning rate, we conducted a comparative experiment on GPT-2 (Small) using the CyclicLR schedule provided in PyTorch with the triangular2 policy [3], training for 100K steps. The test results and their comparisons with CosineAnnealingLR are summarized in the table 10.

Table 10: **Performance of staleness gap penalty on CyclicLR.** We pre-train GPT-2 (Small) with 100K steps for ablation. Both train and validation perplexity results indicate that staleness gap penalty is effective on CyclicLR schedule.

| Method | Staleness Gap Penalty | Train PPL | Validation PPL |
|---|---|---|---|
| CO2 with CosineAnnealingLR | No | 7.52 | 7.56 |
| CO2 with CosineAnnealingLR | Yes | 7.36 | 7.39 |
| CO2 with CyclicLR | No | 7.58 | 7.63 |
| CO2 with CyclicLR | Yes | 7.45 | 7.51 |

The experimental outcomes consistently affirm the favorable impact of the staleness gap penalty technique on training performance, regardless of the choice between CosineAnnealingLR and CyclicLR. While the enhancements in PPL may not be substantial, they exhibit a reliable positive trend. It is noteworthy that PPL values using CyclicLR surpass those using CosineAnnealingLR, potentially owing to sub-optimal learning rate tuning. Despite the cyclic nature of the learning rate in CyclicLR, the staleness gap penalty continues to exert its positive influence. This could be attributed to the periodic decrease in the learning rate during CyclicLR's training iterations, facilitating the efficacy of the

---

[3]https://github.com/bckenstler/CLR

staleness gap penalty. It is pertinent to acknowledge that both tested schedules incorporate a learning rate warm-up during the initial 4000 iterations, which also represents a form of increasing learning rate. From a technical standpoint, we posit that there may exist specific learning rate schedules where the staleness gap penalty technique may not exhibit optimal performance. In such instances, we advocate exploring alternative decaying learning rate schedules or considering the option to disable the staleness gap penalty.

## A.4 CONVERGENCE PROOF

The convergence proof is conducted for a constant learning rate, i.e., $\gamma_t = \gamma$ and constant staleness gap, i.e., $\lambda = 1/\Lambda_t$, and we ignore the outer momentum clipping operation to make the analysis easy to follow. Besides, we follows the standard assumptions (Wang et al., 2019) as below:

**Assumption A.1** *For some existed $L > 0$, there is $\|\nabla f_i(x) - \nabla f_i(y)\| \leq L\|x - y\|$, for all inputs $x, y \in \mathbb{R}^n$ and $i \in \{1, 2, \cdots, G\}$.*

**Assumption A.2** *For all $i \in \{1, 2, \ldots, G\}$, there exists a finite positive constant $\sigma^2$ such that $\mathbb{E}_{\zeta \sim D_i} \left\| \nabla L^{(i)}(\boldsymbol{x}; \zeta) - \nabla f_i(\boldsymbol{x}) \right\|^2 \leq \sigma^2$, i.e., the variance of $f_i(\boldsymbol{x})$ is bounded.*

**Assumption A.3** *There exists a finite positive constant $V$ such that $\mathbb{E} \left\| \boldsymbol{g}_{t,k} - \mathbb{E}\left[\boldsymbol{g}_{t,k}\right] \right\|^2 \leq V$.*

Under these assumptions, the momentum update rule in Algorithm 1 becomes:

$$\boldsymbol{m}_t = \beta \boldsymbol{m}_{t-1} + \lambda(\boldsymbol{x}_{t-1,0} - \boldsymbol{x}_{t-1,\tau}). \tag{5}$$

Let us define:

$$\boldsymbol{g}_{t,k} = \frac{1}{G} \sum_{i=1}^{G} \boldsymbol{g}_{t,k}^{(i)}. \tag{6}$$

Note that the local update rule is:

$$\boldsymbol{x}_{t,k+1}^{(i)} = \boldsymbol{x}_{t,k}^{(i)} - \gamma \boldsymbol{g}_{t,k}^{(i)}. \tag{7}$$

So we have:

$$
\begin{aligned}
\boldsymbol{x}_{t,k+1} &= \frac{1}{G} \sum_{i=1}^{G} \boldsymbol{x}_{t,k+1}^{(i)} \\
&= \frac{1}{G} \left( \sum_i \boldsymbol{x}_{t,k}^{(i)} - \sum_i \gamma \boldsymbol{g}_{t,k}^{(i)} \right) \\
&= \boldsymbol{x}_{t,k} - \gamma \boldsymbol{g}_{t,k} \\
&= \boldsymbol{x}_{t,k-1} - \gamma(\boldsymbol{g}_{t,k-1} + \boldsymbol{g}_{t,k}) \\
&= \boldsymbol{x}_{t,0} - \gamma \sum_{j=0}^{k} \boldsymbol{g}_{t,j}.
\end{aligned}
\tag{8}
$$

The momentum update rules then becomes:

$$\boldsymbol{m}_t = \beta \boldsymbol{m}_{t-1} + \lambda(\boldsymbol{x}_{t-1,0} - \boldsymbol{x}_{t-1,\tau}) = \beta \boldsymbol{m}_{t-1} + \lambda\gamma \sum_{k=0}^{\tau-1} \boldsymbol{g}_{t-1,k}. \tag{9}$$

The outer update rule becomes:

$$
\begin{aligned}
\boldsymbol{x}_{t+1,0} &= \boldsymbol{x}_{t,0} - \alpha \boldsymbol{m}_t \\
&= \boldsymbol{x}_{t,0} - \alpha\beta \boldsymbol{m}_{t-1} - \alpha\lambda\gamma \sum_{k=0}^{\tau-1} \boldsymbol{g}_{t-1,k} \\
&= \boldsymbol{x}_{t,0} - \alpha\lambda\gamma \sum_{k=0}^{\tau-1} \boldsymbol{g}_{t-1,k} + \beta(\boldsymbol{x}_{t,0} - \boldsymbol{x}_{t-1,0}).
\end{aligned}
\tag{10}
$$

We define:

$$\boldsymbol{y}_{t,0} = \boldsymbol{x}_{t,0} + \frac{\beta}{1-\beta} \left( \boldsymbol{x}_{t,0} - \boldsymbol{x}_{t-1,0} \right). \tag{11}$$

So we have:

$$
\begin{aligned}
\boldsymbol{y}_{t+1,0} - \boldsymbol{y}_{t,0} &= \boldsymbol{x}_{t+1,0} - \boldsymbol{x}_{t,0} + \frac{\beta}{1-\beta} \left( \boldsymbol{x}_{t+1,0} - \boldsymbol{x}_{t,0} \right) - \frac{\beta}{1-\beta} \left( \boldsymbol{x}_{t,0} - \boldsymbol{x}_{t-1,0} \right) \\
&= \frac{1}{1-\beta} \left( \boldsymbol{x}_{t+1,0} - \boldsymbol{x}_{t,0} - \beta \left( \boldsymbol{x}_{t,0} - \boldsymbol{x}_{t-1,0} \right) \right) \\
&= -\frac{\alpha\lambda\gamma}{1-\beta} \sum_{k=0}^{\tau-1} \boldsymbol{g}_{t-1,k}.
\end{aligned}
\tag{12}
$$

We then define $\boldsymbol{y}_{t,\tau} = \boldsymbol{y}_{t+1,0}$ and

$$\boldsymbol{y}_{t,k+1} - \boldsymbol{y}_{t,k} = -\frac{\alpha\lambda\gamma}{1-\beta} \boldsymbol{g}_{t-1,k} \triangleq -\frac{\bar{\lambda}\gamma}{1-\beta} \boldsymbol{g}_{t-1,k}. \tag{13}$$

Since $f_i$ is $L$-Smooth, $f = \frac{1}{G} \sum_{i=1}^{G} f_i$ is also $L$-Smooth. then according to $L$-Smooth of $f$ and the Descent Lemma in Bauschke et al. (2017), we have

$$
\begin{aligned}
f(\boldsymbol{y}_{t,k+1}) &= f \left( \boldsymbol{y}_{t,k} - \frac{\bar{\lambda}\gamma}{1-\beta} \, \boldsymbol{g}_{t-1,k} \right) \\
&\leq f(\boldsymbol{y}_{t,k}) - \frac{\bar{\lambda}\gamma}{1-\beta} \left\langle \nabla f\left(\boldsymbol{y}_{t,k}\right), \boldsymbol{g}_{t-1,k} \right\rangle + \frac{\bar{\lambda}^2\gamma^2 L}{2(1-\beta)^2} \|\boldsymbol{g}_{t-1,k}\|^2.
\end{aligned}
\tag{14}
$$

Taking expectation, then we have

$$\mathbb{E}[f(\boldsymbol{y}_{t,k+1})] \leq f(\boldsymbol{y}_{t,k}) - \frac{\lambda\gamma}{1-\beta} \left\langle \nabla f\left(\boldsymbol{y}_{t,k}\right), \mathbb{E}[\boldsymbol{g}_{t-1,k}] \right\rangle + \frac{\bar{\lambda}^2\gamma^2 L}{2(1-\beta)^2} \mathbb{E}\left[\|\boldsymbol{g}_{t-1,k}\|^2\right] \tag{15}$$

According to (15), and we define $f_{\inf} = \inf_x f(\boldsymbol{x})$ (Wang et al., 2019). We have:

$$\frac{1}{T\tau}\sum_{t=0}^{T-1}\sum_{k=0}^{\tau-1}\mathbb{E}\left\|\nabla f\left(\boldsymbol{x}_{t,k}\right)\right\|^2$$

$$\leq \frac{2\left(f\left(\boldsymbol{x}_{0,0}\right)-f_{\inf}\right)+mVL}{\sqrt{GT\tau}}+\frac{1}{T\tau}\sum_{t=0}^{T-1}\sum_{k=0}^{\tau-1}\mathbb{E}\left\|\nabla f\left(\boldsymbol{x}_{t,k}\right)-\mathbb{E}_{t,k}\left[\boldsymbol{g}_{t-1,k}\right]\right\|^2$$

$$+\frac{4GVL^2(\tau-1)}{T\tau}\left(\frac{1-\beta}{\bar{\lambda}}-1\right)^2+\frac{8GVL^2\tau}{T\tau}\frac{\beta^2}{(1-\beta^2)}$$

$$\leq \frac{2\left(f\left(\boldsymbol{x}_{0,0}\right)-f_{\inf}\right)+mVL}{\sqrt{GT\tau}}$$

$$+\frac{1}{T\tau}\sum_{t=0}^{T-1}\sum_{k=0}^{\tau-1}\mathbb{E}\left\|\nabla f\left(\boldsymbol{x}_{t,k}\right)-\mathbb{E}_{t,k}\left[\boldsymbol{g}_{t,k}\right]+\mathbb{E}_{t,k}\left[\boldsymbol{g}_{t,k}\right]-\mathbb{E}_{t,k}\left[\boldsymbol{g}_{t-1,k}\right]\right\|^2$$

$$+\frac{4GVL^2(\tau-1)}{T\tau}\left(\frac{1-\beta}{\bar{\lambda}}-1\right)^2+\frac{8GVL^2\tau}{T\tau}\frac{\beta^2}{(1-\beta^2)}$$

$$\leq \frac{2\left(f\left(\boldsymbol{x}_{0,0}\right)-f_{\inf}\right)+mVL}{\sqrt{GT\tau}} \tag{16}$$

$$+\underbrace{\frac{1}{T\tau}\sum_{t=0}^{T-1}\sum_{k=0}^{\tau-1}\mathbb{E}\left\|\nabla f\left(\boldsymbol{x}_{t,k}\right)-\mathbb{E}_{t,k}\left[\boldsymbol{g}_{t,k}\right]\right\|^2}_{\text{Effect of Local SGD}}$$

$$+\underbrace{\frac{2}{T\tau}\sum_{t=0}^{T-1}\sum_{k=0}^{\tau-1}\mathbb{E}\left[\left(\nabla f\left(\boldsymbol{x}_{t,k}\right)-\mathbb{E}_{t,k}\left[\boldsymbol{g}_{t,k}\right]\right)^\top\left(\mathbb{E}_{t,k}\left[\boldsymbol{g}_{t,k}\right]-\mathbb{E}_{t,k}\left[\boldsymbol{g}_{t-1,k}\right]\right)\right]}_{\text{Effect of cross term}}$$

$$+\underbrace{\frac{1}{T\tau}\sum_{t=0}^{T-1}\sum_{k=0}^{\tau-1}\mathbb{E}\left\|\mathbb{E}_{t,k}\left[\boldsymbol{g}_{t,k}\right]-\mathbb{E}_{t,k}\left[\boldsymbol{g}_{t-1,k}\right]\right\|^2}_{\text{Effect of asynchronous updates}}$$

$$+\underbrace{\frac{4GVL^2(\tau-1)}{T\tau}\left(\frac{1-\beta}{\bar{\lambda}}-1\right)^2+\frac{8GVL^2\tau}{T\tau}\frac{\beta^2}{(1-\beta^2)}}_{\text{Effect of outer momentum}}.$$

Note that $\mathbb{E}_{t,k}\left[\boldsymbol{g}_{t,k}\right]=\mathbb{E}_{t,k}\left[\boldsymbol{g}_{t-1,k}\right]$, so the cross-term and asynchronous-term eliminate, which results:

$$\frac{1}{T\tau}\sum_{t=0}^{T-1}\sum_{k=0}^{\tau-1}\mathbb{E}\left\|\nabla f\left(\boldsymbol{x}_{t,k}\right)\right\|^2 \leq \frac{2\left(f\left(\boldsymbol{x}_{0,0}\right)-f_{\inf}\right)+mVL}{\sqrt{GT\tau}}$$

$$+\underbrace{\frac{1}{T\tau}\sum_{t=0}^{T-1}\sum_{k=0}^{\tau-1}\mathbb{E}\left\|\nabla f\left(\boldsymbol{x}_{t,k}\right)-\mathbb{E}_{t,k}\left[\boldsymbol{d}_{t,k}\right]\right\|^2}_{\text{Effect of Local SGD}} \tag{17}$$

$$+\underbrace{\frac{4GVL^2(\tau-1)}{T\tau}\left(\frac{1-\beta}{\bar{\lambda}}-1\right)^2+\frac{8GVL^2\tau}{T\tau}\frac{\beta^2}{(1-\beta^2)}}_{\text{Effect of outer momentum}}$$

Then, we finally have:

$$\frac{1}{T\tau}\sum_{t=0}^{T-1}\sum_{k=0}^{\tau-1}\mathbb{E}\left\|\nabla f\left(\boldsymbol{x}_{t,k}\right)\right\|^2 = \mathcal{O}\left(\frac{1}{\sqrt{GT\tau}}\right)+\mathcal{O}\left(\frac{G\tau}{T}\right). \tag{18}$$

# B DISCUSSIONS

## B.1 DISTINCTIONS WITH EXISTED METHODS

The stale-synchronous parallel (SSP) framework (Ho et al., 2013; Tang et al., 2020) aims to mitigate the straggler problem with relaxed synchronization. Specifically, SSP allows faster workers to perform more updates than slower ones, reducing the waiting time of faster workers. However, to maintain model consistency and ensure convergence, SSP imposes a staleness bounded barrier that limits the iteration gap between the fastest and slowest workers. Comparing with CO2, their differences mainly lie in these three aspects: 1) The SSP framework is built upon the Parameter-Server architecture. While CO2 as well as the Local-SGD and SlowMo algorithms investigated in the paper all based on the All-Reduce architecture. 2) The SSP-like algorithms indeed introduces asynchronization into its training procedure, however, it does not allow the communication to be overlapped with computations. And its staleness is not punished, which will leads unsatisfactory convergence performances. As comparison, the asynchronous communication in CO2 makes the communication can be overlapped with multiple-step local updates, which hugely improves the training efficiency. And the staleness gap penalty technique allows to mitigate the negative effects on convergence results. These imply the scaling efficiency and convergence performance of SSP-like algorithms will be worse than CO2 in practice. 3) The convergence analysis of SSP framework indicates that the convergence rate of SSP in normal cases is $O(\frac{1}{\sqrt{T}})$. Comparing with CO2, when the total steps $T\tau$ is sufficiently large, i.e, $T \gg G^3\tau^3$, the convergence rate of CO2 is $\mathcal{O}\left(\frac{1}{\sqrt{GT\tau}}\right)$. So theoretically, the convergence rates of these two algorithms are similar.

The Cooperative SGD (Wang & Joshi, 2021) aims to unify a variety of local-update SGD algorithms, such as local SGD, elastic averaging SGD, and decentralized parallel SGD, under the proposed unified Cooperative SGD framework. It also provides a unified convergence analysis for the methods under this framework. As one of the by-products, it proposes to add an auxiliary variable $z$ on the elastic averaging SGD, which serves as a local copy of the model. It can be seen as a preliminary approach that enables the asynchronous communication of model averaging to overlap with computation. However, our method leverages more sophisticated asynchronous communication design to improve the convergence. The differences between our method and the Cooperative SGD are threefold: 1) The CO2 algorithm is built upon SlowMo, which involves slow momentum and outer updates to improve the convergence; 2) CO2 leverages more sophisticated asynchronous communication design together with the outer momentum and outer updates to hide more communications; 3) CO2 also introduces staleness gap penalty and outer momentum clipping to improve the convergence as well as training stability.

The Overlap-Local SGD (Wang et al., 2020) method is also able to overlap communication by local computations via adding an anchor model on each node. The distinctions between CO2 and Overlap-Local-SGD are threefold: 1) Overlap-Local-SGD achieves communication-computation overlap on the top of naive Local-SGD, whose convergence performance is normally not good enough. On the other hand, the CO2 is built on SlowMo, which involves outer loop includes outer momentum and outer updates to improve the convergence of native Local-SGD. 2) As mentioned by the reviewer, CO2 introduced staleness gap penalty and outer momentum clipping to prevent divergence and improve training stability, which has been verified effective via extensive experiments. 3) The Overlap-Local-SGD is only tested on the CIFAR-10 image classification task using ResNet-18, while CO2 has been widely verified on multiple tasks on both CV and NLP fields, including image classification, semantic segmentation, 3D point cloud reconstruction, autoregressive language modeling and bidirectional language modeling.

## B.2 MEMORY OVERHEADS OF CO2

As an asynchronous method, CO2 leverages the advantages of a high overlap ratio facilitated by asynchronous communication, while contending with an augmented memory footprint due to the one-step asynchronous delay. Here we present a preliminary comparative analysis of the memory footprint of the methods discussed in the paper, using the widely adopted Adamw optimizer as the baseline and excluding considerations of mixed precision. Local-Adamw, a variant of Adamw, conducts local forward and backward passes and updates parameters without synchronizing gradients. It shares the same memory footprint as Adamw but exhibits lower communication frequency and

diminished accuracy performance. SlowMo, can be seen as a variant of Local-Adamw, introduces slow momentum calculation and outer model updates to enhance accuracy. Despite maintaining the same communication frequency as Local-Adamw, SlowMo incurs an additional memory footprint (twice that of the model itself) due to the introduced slow momentum and outer updates. In comparison, $CO_2$ introduces a one-step-delay outer momentum and outer model updates to overlap communication with multiple steps of local-updating computation, enhancing communication-computation overlap. This improvement comes at the cost of an additional memory footprint (twice that of the model itself) compared to SlowMo, attributed to asynchronous operations. In summary, SlowMo requires additional memory footprint which corresponds to 2 model copies than Local-Adamw. Then on the top of SlowMo, $CO_2$ involves another additional memory footprint which corresponds to 2 model copies, so totally, $CO_2$ needs additional memory overheads of 4 model copies comparing with the vanilla Adamw or Local-Adamw.

Despite the heightened memory overhead introduced by $CO_2$, it enables full overlap (in the best case) of communication and computation, particularly advantageous for large clusters with limited inter-node connections. Techniques such as ZeRO-1, ZeRO-2, and ZeRO-3 have been incorporated to alleviate redundant memory footprint issues associated with optimizer states, gradients, and model parameters within the inner loop training of $CO_2$. These measures could mitigate the adverse effects of $CO_2$'s augmented memory overheads.

### B.3 COMPATIBILITY WITH PARALLELISM TECHNOLOGIES

Given the additional memory footprints associated with asynchronous updates, the compatibility of $CO_2$ with various parallelism techniques becomes an indispensable consideration, especially in the context of large-scale model training. During the implementation of the algorithm, we systematically investigated its compatibility with ZeRO-series optimizers. Currently, ZeRO-1, ZeRO-2, and ZeRO-3 are integrated into the inner loop training of $CO_2$, with the exclusion of the outer loop. We conducted thorough experiments to validate the effectiveness and efficiency of these integrations. In particular, with respect to ZeRO-3, which entails more sophisticated communication during both forward and backward passes, we had considered its compatibility with $CO_2$. Given $CO_2$'s local-updating behavior in the inner loop, continuous gradient synchronization is deemed unnecessary at each step. As a result, communication operations for $CO_2$ with ZeRO-3 involve only two all-gather collective communication primitives at the beginning of the forward and backward passes, respectively, which differs from the original implementation of ZeRO-3.

Concerning tensor parallelism, sequence parallelism, and their integration with $CO_2$, while we have not implemented them at this moment, their mechanisms appear non-conflicting with $CO_2$'s inner loop training. Tensor parallelism divides large matrix multiplications across multiple devices, and sequence parallelism divides the input sequence into chunks, processing them simultaneously on different devices for efficient training of long sequences. We will try to integrate them with $CO_2$ in the future work.

