# OpenReview forum: "CO2: Efficient Distributed Training with Full Communication-Computation Overlap"
_ICLR.cc/2024/Conference — ICLR 2024 spotlight_

### Official Review · Reviewer_Qwcz · 2023-10-30

**Soundness:** 4 excellent
**Presentation:** 4 excellent
**Contribution:** 3 good
**Rating:** 8
**Confidence:** 2

**Summary:**

The paper proposes CO2, a new approach that enables efficient distributed training of large language models on clusters with limited bandwidth. CO2 introduces local-updating and asynchronous communication to the distributed data-parallel training, allowing for full overlap of communication with computation. The approach achieves 100% scalability even on clusters with limited communication bandwidth. The paper also introduces staleness gap penalty and outer momentum clipping techniques to improve convergence and training stability. The proposed approach is validated through extensive experiments on computer vision and natural language processing tasks as well.

**Strengths:**

+ The paper is well-written and comprehensible.
+ The code is available in this work.
+ The utilization of local updating and asynchronous communication makes a full overlap of computation and communication.
+ The paper provides enough theoretical explainability and empirical validation.
+ The experimental results are sound and promising.

**Weaknesses:**

I do not have much to comment on the weakness, as this work goes beyond my acceptance threshold.

**Questions:**

How many runs for each task? I understand that training a Language Learning Model from scratch can be quite costly. However, conducting the experiment only once may not yield persuasive results.

---

> ### Author Response · Authors · 2023-11-22
> **Response to Reviewer Qwcz**
>
> Dear reviewer,
>
> Thanks for the comments and questions. Please see below for our response to your concern.
>
> **Q1: How many runs for each task? Conducting the experiment only once may not yield persuasive results**
>
> A1:
> We thank for the helpful comments. All runs in the convergence experiments were executed five times with distinct random seeds to ensure robust results. And for the throughput experiments, the presented throughput results are averaged values extracted from iterations 100 to 200 for each experiment, which is in order to reduce fluctuations. We will emphasize these experiment details in the forthcoming rebuttal revision version of the paper.

---

### Official Review · Reviewer_rgZH · 2023-11-04

**Soundness:** 2 fair
**Presentation:** 3 good
**Contribution:** 3 good
**Rating:** 6
**Confidence:** 4

**Summary:**

The paper proposes an approach called CO2 to improve throughput of distributed model training by overlapping computation and communication. Building on prior work that perform multiple training iterations with local updates before each global model synchronization, CO2 enables further throughput improvement by making the global synchronization asynchronous and overlapped with the next round of local updates. CO2 proposes two techniques for addressing the convergence issues of the asynchrony: (i) staleness penalty gap, and (ii) outer momentum clipping. The paper presents theoretical analyses of the convergence guarantees of these two techniques. The evaluation results show that CO2 can achieve convergence results comparable to baselines that are fully synchronous (e.g, Adam) and better than those using local updates (e.g, LocalSGD). The experimental results also show the throughput and scalability of CO2 are better than Adam.

**Strengths:**

The paper is tackling an important problem since communication is a major bottleneck for scaling model sizes and training hardware, and so approaches for reducing communication overheads are very relevant to the community.

The idea of overlapping communication with computation is reasonable given the cost-effectiveness. I also liked the fact that the paper attempts to quantify and fix the resulting staleness update problem.

The evaluation considers a diverse and important set of workloads and hardware environments, which helps to understand the generality of CO2.

**Weaknesses:**

I observe some critical problems in the draft that raise the question of whether CO2 can simultaneously achieve good convergence and high throughput.

1. The convergence and throughput trade-off of inner loop step count ($\tau$) is not clearly reported in evaluation. In particular, the convergence results in Tables 1 & 2 should include the corresponding $\tau$ and throughput. I was unable to determine whether the good convergence results are achieved with $\tau$ that also provides throughput benefits.

2. The paper is silent on the memory overheads of CO2 relative to baselines, even though Algorithm 2 suggests that multiple copies of the model is required to support asynchronous communication.

3. Equation 3 assumes learning rate decay in the inner loop which is not true for learning rate schedules, such as cyclic, which involve learning rate increases.

4. It is unclear to me whether CO2 can achieve expected throughput benefits in scenarios with parallelism techniques (e.g., tensor slicing, sequence parallelism, and zero stage 3) that introduce communication to forward/backward passes. It seems these (synchronous) communication operations would interfere with the overlapped communication and hurt overall throughput. Evaluating such scenarios could help to better understand the generality of CO2.

**Questions:**

See weaknesses.

---

> ### Author Response · Authors · 2023-11-20
> **Response to Reviewer rgZH (Part 1)**
>
> Dear reviewer,
>
> Thank you for your valuable and constructive feedback, which contributes a lot to the enhancement of our paper. Below, we provide our responses addressing your concerns. And all the revisions will be updated in the forthcoming rebuttal revision version of the paper.
>
> **Q1: The trade-off between convergence and throughput benefits**
>
> A1:
> Thanks for the helpful comments. The omission of throughput results in Table 1 and 2 stems from (part of) our convergence experiments being conducted on 8 A100 nodes connected through a high-speed RoCE RDMA network. And the investigated model sizes were relatively small, which mitigates the discernibility of CO2's speed advantage. As rightly pointed out by you and other reviewers, we recognize the significance of showing the trade-off between convergence and throughput benefits for CO2, we have addressed this gap by providing the missing throughput data in the below Tables. The presented throughput results represent averaged values extracted from iterations 100 to 200 for each experiment. Given the extensive nature of our experiments, convergence outcomes in Table 1 and 2 originate from two distinct clusters. Specifically, the IC and ALM tasks were executed on an A100 cluster equipped with RoCE RDMA high-speed inter-node connections, featuring 8 nodes and a total of 64 GPUs. Conversely, the SS, PC, and BLM tasks were conducted on a 3090 cluster with a standard TCP/IP Ethernet inter-node connection, comprising 4 nodes and 32 GPUs. Upon observing tasks trained on the A100 platform, it is evident that the throughput of CO2 surpasses other counterparts, although with a slight advantage. This advantage becomes more pronounced on tasks trained on the 3090 platform. It is reasonable to infer that this advantage will be amplified on larger clusters, especially those with slower inter-node connections.
>
> Prior to commencing large-scale convergence experiments, we meticulously tune the values of $\tau$ for each task using a simple grid search strategy within the range of {1, 3, 6, 12, 24, 48, 96, 192} to reach a balance of accuracy and throughput. Hyperparameter tuning experiments are conducted on smaller versions of models when the model to be trained is large, owing to the associated high time and resource costs. We start the tuning of $\tau$ from 1 to larger candidate values given the consideration of high-speed communication on the corresponding platform. It is worth noting that, in addition to CO2, other methods such as Local-SGD/Adamw and SlowMo also require tuning of $\tau$. In practice, we exclusively tune $\tau$ for CO2 and employ the same tuned $\tau$ values for Local-SGD/Adamw and SlowMo. This is because of the similar role played by $\tau$ in these local-updating methods. Employing identical $\tau$ values for different methods on the same task ensures a fair comparison of their convergence and training throughput.
>
> | Task | Model| SGD (Adamw) | Local-SGD (Local-Adamw) | SlowMo     | CO2 |
> |-|-|-|-|-|-|
> |      |             | Acc / Thpt  | Acc / Thpt / $\tau$             | Acc / Thpt / $\tau$ | Acc / Thpt / $\tau$ |
> | IC (A100)    | ResNet-50   | 76.92 / 108739      | 75.57 / 108758 / 1                  | 77.12 / 108741 / 1       | 77.14 / 108753 / 1       |
> | IC (A100)   | ViT (Base)  | 81.33 / 39422        | 78.43 / 39512 / 3                   | 79.83 / 39509 / 3       | 80.95 / 39533 / 3       |
> | IC (A100)    | VVT (Large) | 83.64 / 44375        | 81.09 / 44390 / 1                   | 82.75 / 44376 / 1       | 83.38 / 44387 / 1       |
> | SS (3090)   | VVT (Large) | 47.82 / 5384        | 44.25 / 5528 / 6                   | 47.51 / 5521 / 6       | 47.80 / 5562 / 6       |
> | PC (3090)   | Point-MAE   | 68.56 / 5859        | 64.25 / 5931 / 3                   | 68.69 / 5904 / 3       | 68.89 / 5956 / 3       |
>
> \* For IC and SS tasks, we present the throughput results in "images/s", the image resolution is 224x224; For PC task, we present the throughput results in "point clouds/s", each point cloud has 1024 points.
>
> | Task | Model| Adamw      | Local-Adamw | SlowMo     | CO2|
> |-|-|-|-|-|-|
> |      |                 | Acc / Thpt | Acc / Thpt / $\tau$  | Acc / Thpt / $\tau$ | Acc / Thpt / $\tau$ |
> | ALM (A100)  | GPT-2 (Small)   | 7.44 / 6.543e6       | 7.95 / 6.556e6 / 3        | 7.34 / 6.554e6 / 3       | 7.37 / 6.556e6 / 3       |
> | ALM (A100)  | GPT-2 (Medium)  | 6.61 / 2.084e6       | 7.49 / 2.094e6 / 3       | 6.41 / 2.091e6 / 3       | 6.36 / 2.092e6 / 3       |
> | ALM (A100)  | GPT-2 (Large)   | 6.26 / 1.052e6       | 7.00 / 1.059e6 / 6       | 6.29 / 1.053e6 / 6       | 6.13 / 1.056e6 / 6      |
> | ALM (A100)  | TN-LLM (7B)     | 16.82 / 0.281e6      | 18.63 / 0.303e6 / 12       | 16.95 / 0.301e6 / 12      | 16.78 / 0.308e6 / 12      |
> | BLM (3090)  | RoBERTa (Large) | 3.96 / 2262       | 4.38 / 2815 / 6        | 3.98 / 2794 / 6       | 3.95 / 2892 / 6       |
>
> \* For ALM and BLM tasks, we present the throughput results in "tokens/s".

---

> ### Author Response · Authors · 2023-11-20
> **Response to Reviewer rgZH (Part 2)**
>
> **Q2: The memory overheads of CO2**
>
> A2:
> Thank you for the comprehensive review. As an asynchronous method, CO2 leverages the advantages of a high overlap ratio facilitated by asynchronous communication, while contending with an augmented memory footprint due to the one-step asynchronous delay. Here we present a preliminary comparative analysis of the memory footprint of the methods discussed in the paper, using the widely adopted Adamw optimizer as the baseline and excluding considerations of mixed precision. Local-Adamw, a variant of Adamw, conducts local forward and backward passes and updates parameters without synchronizing gradients. It shares the same memory footprint as Adamw but exhibits lower communication frequency and diminished accuracy performance. SlowMo, can be seen as a variant of Local-Adamw, introduces slow momentum calculation and outer model updates to enhance accuracy. Despite maintaining the same communication frequency as Local-Adamw, SlowMo incurs an additional memory footprint (twice that of the model itself) due to the introduced slow momentum and outer updates. In comparison, CO2 introduces a one-step-delay outer momentum and outer model updates to overlap communication with multiple steps of local-updating computation, enhancing communication-computation overlap. This improvement comes at the cost of an additional memory footprint (twice that of the model itself) compared to SlowMo, attributed to asynchronous operations.
>
> Despite the heightened memory overhead introduced by CO2, it enables full overlap (in the best case) of communication and computation, particularly advantageous for large clusters with limited inter-node connections. Techniques such as ZeRO-1, ZeRO-2, and ZeRO-3 have been incorporated to alleviate redundant memory footprint issues associated with optimizer states, gradients, and model parameters within the inner loop training of CO2. These measures could mitigate the adverse effects of CO2's augmented memory overheads. Furthermore, some ongoing efforts are dedicated to refining the CO2 algorithm in subsequent versions, with a focus on mitigating additional memory overheads.
>
> **Q3: The non-decay learning rate consideration**
>
> A3:
> Thank you for bringing up this point. Indeed, the assumption of a decaying learning rate when designing the staleness gap may be considered somewhat restrictive. We acknowledge that our experiments did not thoroughly explore various types of learning rate schedules. To address this concern, we conducted a comparative experiment on GPT-2 (Small) using the CyclicLR schedule provided in PyTorch with the "triangular2" policy (see [pytorch link](https://pytorch.org/docs/stable/generated/torch.optim.lr_scheduler.CyclicLR.html) and [github link](https://github.com/bckenstler/CLR)), training for 100K steps. The test results and their comparisons with CosineAnnealingLR are summarized in the table below:
>
> | Experiments             |Train PPL      |Validation PPL |
> |-------------------------|---------------|---------------|
> | CO2, CosineAnnealingLR, No Penalty |7.52           |7.56           |
> | CO2, CosineAnnealingLR, Penalty    |7.36           |7.39           |
> | CO2, CyclicLR, No Penalty |7.58           |7.63           |
> | CO2, CyclicLR, Penalty    |7.45           |7.51           |
>
> The experimental outcomes consistently affirm the favorable impact of the staleness gap penalty technique on training performance, regardless of the choice between CosineAnnealingLR and CyclicLR. While the enhancements in perplexity (PPL) may not be substantial, they exhibit a reliable positive trend. It is noteworthy that PPL values using CyclicLR surpass those using CosineAnnealingLR, potentially owing to suboptimal learning rate tuning. Despite the cyclic nature of the learning rate in CyclicLR, the staleness gap penalty continues to exert its positive influence. This could be attributed to the periodic decrease in the learning rate during CyclicLR's training iterations, facilitating the efficacy of the staleness gap penalty. It is pertinent to acknowledge that both tested schedules incorporate a learning rate warm-up during the initial 4000 iterations, which also represents a form of increasing learning rate. From a technical standpoint, we posit that there may exist specific learning rate schedules where the staleness gap penalty technique may not exhibit optimal performance. In such instances, we advocate exploring alternative decaying learning rate schedules or considering the option to disable the staleness gap penalty.

---

> > ### Comment · Reviewer_rgZH · 2023-11-22
> > **Memory overheads of CO2**
> >
> > Thanks for your response. While it mostly addresses my concerns, it did not clarify how many model copies are required by CO2, especially with reference to Algorithm 2 as I pointed out my comment. Your response seems to suggest that 2 copies are required (similar to SlowMo) but I think Algorithm 2 suggests at least 4. It is very possible that I have overcounted in Algorithm 2, but that is why addressing this question directly in the paper would help readers like me avoid such confusion. I appreciate that leveraging ZeRO is a reasonable mitigation to the memory overheads.

---

> > > ### Author Response · Authors · 2023-11-23
> > > **Response to Reviewer rgZH on memory overheads of CO2**
> > >
> > > We thank for your meticulous review. Your counting is correct. As have described in above response for Q2, SlowMo requires additional memory footprint which corresponds to 2 model copies than Local-Adamw. Then on the top of SlowMo, CO2 involves another additional memory footprint which corresponds to 2 model copies, so totally, CO2 needs additional memory overheads of 4 model copies comparing with the vanilla Adamw or Local-Adamw. To alleviate any potential confusion among readers, we will integrate all these details into the paper.

---

> ### Author Response · Authors · 2023-11-20
> **Response to Reviewer rgZH (Part 3)**
>
> **Q4: The compatibility with parallelism technologies**
>
> A4:
> We appreciate the comprehensive assessment provided. Given the additional memory footprints associated with asynchronous updates, the compatibility of CO2 with various parallelism techniques becomes an indispensable consideration, especially in the context of large-scale model training. During the implementation of the algorithm, we systematically investigated its compatibility with ZeRO-series optimizers. Currently, ZeRO-1, ZeRO-2, and ZeRO-3 are integrated into the inner loop training of CO2, with the exclusion of the outer loop. We conducted thorough experiments to validate the effectiveness and efficiency of these integrations. In particular, with respect to ZeRO-3, which entails more sophisticated communication during both forward and backward passes, we had considered its compatibility with CO2. Given CO2's local-updating behavior in the inner loop, continuous gradient synchronization is deemed unnecessary at each step. As a result, communication operations for CO2 with ZeRO-3 involve only two all-gather collective communication primitives at the beginning of the forward and backward passes, respectively, which differs from the original implementation of ZeRO-3.
>
> Concerning tensor parallelism, sequence parallelism, and their integration with CO2, while we have not implemented them at this moment, their mechanisms appear non-conflicting with CO2's inner loop training. Tensor parallelism divides large matrix multiplications across multiple devices, and sequence parallelism divides the input sequence into chunks, processing them simultaneously on different devices for efficient training of long sequences. We will try to integrate them with CO2 in the future work.

---

> > ### Comment · Reviewer_rgZH · 2023-11-22
> > **Appreciate the response**
> >
> > I greatly appreciate the careful and detailed response to my questions. Overall, I found the responses valuable to improving my understanding and rating of the work. I will raise my score to 6.

---

### Official Review · Reviewer_m91N · 2023-11-08

**Soundness:** 3 good
**Presentation:** 3 good
**Contribution:** 3 good
**Rating:** 6
**Confidence:** 4

**Summary:**

To address the communication problem in large-scale distributed training of deep neural networks, the paper proposes a combination of local-SGD and asynchronous communication to derive a new distributed training algorithm named CO2. In CO2, two novel approaches are developed to ensure that CO2 aligns the convergence performance with conventional distributed data-parallel algorithms. Experiments are conducted on a 64-GPU testbed, showing that CO2 outperforms existing methods significantly. The studied problem is timely and important. The paper is also well-written.

**Strengths:**

- Propose a new distributed training algorithm, CO2, using local updates and asynchronous communication to alleviate the communication problem in conventional synchronous data-parallel distributed training.
- New tricks to address the convergence problem in stale gradients.
- Comphesive experiments to show the effectiveness of CO2.

**Weaknesses:**

- Some stale parallel algorithms (e.g., SSP [ref1]), whose key ideas are quite similar to CO2, were not included in the discussion and comparison. The survey paper [ref2] may help find SSP-like methods for comparison.
- It seems that 100% scaling efficiency is over-claimed. The scaling efficiency highly depends on $\tau$. Higher $\tau$ has better scaling efficiency but has worse convergence performance. Thus, achieving 100% scaling efficiency with a $\tau>1$ while sacrificing the convergence performance cannot conclude the algorithm has true 100% scaling efficiency.

[ref1] More Effective Distributed ML via a Stale Synchronous Parallel Parameter Server, NeurIPS 2013.
[ref2] Communication-efficient distributed deep learning: A comprehensive survey, arXiv 2020.

**Questions:**

- How about comparing with SSP-like algorithms in terms of theoretical convergence bound and empirical scaling efficiency?
- How $\tau$ is set in Table 1?
- How about the end-to-end training performance (i.e., time to accuracy)?
- How to choose $\tau$ in a new distributed GPU cluster?

---

> ### Author Response · Authors · 2023-11-22
> **Response to Reviewer m91N (Part 1)**
>
> Dear reviewer,
>
> Thank you for your valuable and constructive feedback, which contributes a lot to the enhancement of our paper. Below, we provide our responses addressing your concerns. And all the revisions will be updated in the forthcoming rebuttal revision version of the paper.
>
> **Q1: How about comparing with SSP-like algorithms in terms of theoretical convergence bound and empirical scaling efficiency?**
>
> A1:
> Thanks for bringing this up. The stale-synchronous parallel (SSP) framework as described in [ref2] aims to mitigate the straggler problem with relaxed synchronization. Specifically, SSP allows faster workers to perform more updates than slower ones, reducing the waiting time of faster workers. However, to maintain model consistency and ensure convergence, SSP imposes a staleness bounded barrier that limits the iteration gap between the fastest and slowest workers. Comparing with CO2, their differences mainly lie in these three aspects:
> 1) The SSP framework as described in [ref1] and [ref2] is built upon the Parameter-Server architecture. While CO2 as well as the Local-SGD and SlowMo algorithms investigated in the paper all based on the All-Reduce architecture.
> 2) The SSP-like algorithms indeed introduces asynchronization into its training procedure, however, it does not allow the communication to be overlapped with computations. And its staleness is not punished, which will leads unsatisfactory convergence performances. As comparison, the asynchronous communication in CO2 makes the communication can be overlapped with multiple-step local updates, which hugely improves the training efficiency. And the staleness gap penalty technique allows to mitigate the negative effects on convergence results. These imply the scaling efficiency and convergence performance of SSP-like algorithms will be worse than CO2 in practice.
> 3) The convergence analysis section in [ref2] indicates that the convergence rate of SSP in normal cases is $O(\frac{1}{\sqrt{T}})$ . Comparing with CO2, when the total steps $T\tau$ is sufficiently large, i.e, $T\gg G^3\tau^3$, the convergence rate of CO2 is $\mathcal{O}\left(\frac{1}{\sqrt{G T \tau}}\right)$. So theoretically, the convergence rates of these two algorithms are similar.
>
>
> **Q2: 100% scaling efficiency is over-claimed**
>
> A2:
> We appreciate the thoughtful suggestion. Indeed, the original concept behind the design of CO2 aims to achieve a full overlap of the communication with multi-step local computation, the length of which is determined by the parameter $\tau$. However, it is acknowledged that an excessively large $\tau$ may have a detrimental impact on convergence performance, introducing a trade-off between the overlap ratio of communication and computation and the overall convergence. In previous experiments, we have observed that an appropriately chosen $\tau$ can achieve a high level of overlap (100%) while maintaining satisfactory accuracy results. We fully understand the concerns raised and will address them by refining the relevant statements in the paper during the rebuttal revision, aiming for a more balanced depiction of the scalability claims.

---

> ### Author Response · Authors · 2023-11-22
> **Response to Reviewer m91N (Part 2)**
>
> **Q3: How $\tau$ is set in Table 1?**
>
> A3:
> Thanks for the valuable comments. We agree that describing the tuning procedure and the setting of $\tau$ in experiments are helpful for understanding the algorithm. In our experiments, we tune the values of $\tau$ for each task using a simple grid search strategy within the range of {1, 3, 6, 12, 24, 48, 96, 192} to reach a balance of accuracy and throughput. The tuning experiments are conducted on smaller versions of models when the model to be trained is large, owing to the associated high time and resource costs. We start the tuning of $\tau$ from 1 to larger candidate values given the consideration of high-speed communication on the corresponding platform. It is worth noting that, in addition to CO2, other methods such as Local-SGD/Adamw and SlowMo also require tuning of $\tau$. In practice, we exclusively tune $\tau$ for CO2 and employ the same tuned $\tau$ values for Local-SGD/Adamw and SlowMo. This is because of the similar role played by $\tau$ in these local-updating methods. Employing identical $\tau$ values for different methods on the same task ensures a fair comparison of their convergence and training throughput.
>
> We list the values of $\tau$ in all the convergence experiments in the paper as below:
>
> | Task | Model| SGD (Adamw) | Local-SGD (Adamw) | SlowMo     | CO2 |
> |-|-|-|-|-|-|
> |      |             | Acc  | Acc / $\tau$             | Acc / $\tau$ | Acc / $\tau$ |
> | IC (A100)    | ResNet-50   | 76.92      | 75.57 / 1                  | 77.12 / 1       | 77.14 / 1       |
> | IC (A100)   | ViT (Base)  | 81.33        | 78.43 / 3                   | 79.83 / 3       | 80.95 / 3       |
> | IC (A100)    | VVT (Large) | 83.64        | 81.09 / 1                   | 82.75 / 1       | 83.38 / 1       |
> | SS (3090)   | VVT (Large) | 47.82        | 44.25 / 6                   | 47.51 / 6       | 47.80 / 6       |
> | PC (3090)   | Point-MAE   | 68.56        | 64.25 / 3                   | 68.69 / 3       | 68.89 / 3       |
>
> | Task | Model| Adamw      | Local-Adamw | SlowMo     | CO2|
> |-|-|-|-|-|-|
> |      |                 | Acc | Acc / $\tau$  | Acc / $\tau$ | Acc / $\tau$ |
> | ALM (A100)  | GPT-2 (Small)   | 7.44       | 7.95 / 3        | 7.34 / 3       | 7.37 / 3       |
> | ALM (A100)  | GPT-2 (Medium)  | 6.61       | 7.49 / 3       | 6.41 / 3       | 6.36 / 3       |
> | ALM (A100)  | GPT-2 (Large)   | 6.26       | 7.00 / 6       | 6.29 / 6       | 6.13 / 6      |
> | ALM (A100)  | TN-LLM (7B)     | 16.82      | 18.63 / 12       | 16.95 / 12      | 16.78 / 12      |
> | BLM (3090)  | RoBERTa (Large) | 3.96       | 4.38 / 6        | 3.98 / 6       | 3.95 / 6       |
>
>
> **Q4: How about the end-to-end training performance (i.e., time to accuracy)?**
>
> A4:
> Thanks for the valuable question. Actually, we had conducted experiments on GPT-2 (Small, Medium and Large) to show the time to accuracy performances of CO2. The resulted curves are presented in Table 7 in the Appendix of the paper. Note that the experiments are performed on the A100 platform with high-speed inter-node connections. Even with the high-speed connections, CO2 still exhibits the fastest convergence speed on relative time compared to other existing methods.
>
> **Q5: How to choose $\tau$ in a new distributed GPU cluster?**
>
> A5:
> Thank you for your question. We use grid search to select $\tau$. Specifically, we employ a simple grid search strategy within the range of {1, 3, 6, 12, 24, 48, 96, 192} to determine the value of $\tau$, aiming to achieve a balance between accuracy and throughput. To reduce computational demands during hyperparameter tuning for large models, we first conduct experiments on smaller model versions and then apply the hyperparameter to larger models. Similar to CO2, other methods such as Local-SGD/Adamw and SlowMo also need the tuning of $\tau$. In our practical implementation, we first fine-tune $\tau$ for CO2 and subsequently apply the same tuned $\tau$ values for Local-SGD/Adamw and SlowMo. Standardizing $\tau$ values across different methods for a given task ensures a fair and meaningful comparison of their convergence behavior and training throughput.

---

### Official Review · Reviewer_vfwG · 2023-11-09

**Soundness:** 2 fair
**Presentation:** 3 good
**Contribution:** 3 good
**Rating:** 8
**Confidence:** 4

**Summary:**

The paper introduces CO2, a framework for improved communication/computation overlap for distributed deep learning, especially in the case of limited network bandwidth. CO2 leverages local SGD, performing a fixed (tunable) number of local iterations while allreduces perform synchronization in the background, allowing communication to almost always be hidden. To ensure good convergence, CO2 computes a staleness gap metric and uses this to scale updates, as well as a clipping mechanism to limit the variance of updates. A convergence bound is proven and experiments on a variety of network architectures and datasets show convergence matches that of standard SGD and other communication-avoiding algorithms; in the low-bandwidth regime, CO2 additionally offers significantly improved performance and scalability.

**Strengths:**

1. This paper is addressing an important situation: communication-bound training workloads. This can occur due to both large models and slower interconnects. I appreciate that the paper specifically and clearly calls out lower-bandwidth networks as an area it is focused on. While the idea is relatively straightforward, it includes some details to get it to work well in practice.
2. The paper adequately specifies its proposed algorithm and includes some theoretical justification to support its claims.
3. There are extensive experiments on a variety of models, including relatively large ones, demonstrating roughly equivalent convergence curves, indicating that the method does not compromise learning.
4. Scalability studies are also conducted, showing slightly improved performance on high-bandwidth networks and significantly improved performance on low-bandwidth networks relative to a standard allreduce implementation.

**Weaknesses:**

1. I think the claims of "perfect 100% scalability" are a bit oversold. This relies on appropriately selecting $\tau$, the number of local steps between global communications; it seems clear that if you can arbitrarily set the amount of computation done to hide communication, you can easily hide it. (Though I wish to be clear that the paper is clear that you can't make $\tau$ arbitrarily high and still achieve good convergence.) This also neglects other aspects of training which may limit scalability (e.g., I/O for data ingestion).
2. The paper does not provide guidance on selecting an appropriate $\tau$, and in its experiments searches over a small set of potential values. This seems like a challenging parameter to tune in practice, as it could significantly increase hyperparameter tuning costs.
3. It is not clear to me how the paper improves upon existing communication-efficient works which try to tune the communication frequency to achieve both good learning and runtime performance. In particular, works like Wang & Joshi, "Adaptive Communication Strategies to Achieve the Best Error-Runtime Trade-off in Local-Update SGD", or Haddadpour et al., "Local SGD with Periodic Averaging: Tighter Analysis and Adaptive Synchronization", seem like relevant points of comparison.
4. The paper lacks implementation details. Specifically, it does not specify how the asynchronous allreduce is implemented (e.g., is it using a NCCL allreduce on a separate CUDA stream?). It is also not clear whether the asynchronous allreduce is operating on a separate weight/gradient buffer from the one being used for computation; or what the memory overheads of the method are.
5. While I appreciate that the experiments were run multiple times (Section 4.1), the results do not include any measure of variance. This makes it hard to understand whether CO2 amplifies the variance between runs and how much methods actually differ.
6. Scalability is only evaluated on one model. I would be interested to see how models other than the TransNormer-LLM scale; in my experience, smaller models tend to benefit less from communication optimizations as they are already often able to hide most communication.
7. The scaling study in Section 4.3 does not include any comparisons with other communication-efficient methods. Given that SlowMo demonstrates very similar convergence curves, it seems prudent to see whether CO2 offers better scalabiltiy.
8. From a performance perspective, the paper is missing a detailed analysis substantiating its claims. In particular, the communication/computation overlap achieved is never actually measured.

**Questions:**

1. I think the paper would be stronger if the claims of "perfect 100% scalability" were toned down and better contextualized. (See above for some details.)
2. How should $\tau$ be selected? Is hyperparameter tuning the only way to do so?
3. How does the paper improve upon prior works which tune the communication frequency (see above for some references)? Could these approaches be used to tune $\tau$ automatically?
4. Please add implementation details and a discussion of memory overheads. I think memory may be especially relevant for larger models such as LLMs.
5. Please add the observed variance to the accuracy results. It would also be good to include error bars in the scaling performance results.
6. How do other models considered in the paper (e.g., ResNets or ViTs) scale?
7. How do other communication-efficient (e.g., SlowMo) methods scale on the fast and slow network?
8. How much communication/computation overlap is actually achieved by CO2, particularly at scale?
9. A more minor point: The paper refers to gradient bucketing as a way to overlap communication and computation (e.g., in Section 1). I think this is not quite correct; rather, gradient bucketing is a latency/bandwidth tradeoff (performing fewer allreduces on larger buffers). While this can be more efficient, and consequently improve communication/computation overlap, it does not itself enable overlap.

-----

In light of the authors' response and promised updates, I have raised my score. They have addressed a number of points above.

---

> ### Author Response · Authors · 2023-11-22
> **Response to Reviewer vfwG (Part 1)**
>
> Dear reviewer,
>
> We appreciate your valuable and constructive feedback, which significantly contributes to the improvement of our paper. In response to your concerns, we have provided detailed responses, and all the revisions will be incorporated into the upcoming rebuttal revision version of the paper.
>
> **Q1: Tone down and better contextualize the claims of "perfect 100% scalability"**
>
> A1:
> We appreciate the thoughtful suggestion. Indeed, the original concept behind the design of CO2 aims to achieve a full overlap of the communication with multi-step local computation, the length of which is determined by the parameter $\tau$. However, it is acknowledged that an excessively large $\tau$ may have a detrimental impact on convergence performance, introducing a trade-off between the overlap ratio of communication and computation and the overall convergence. In previous experiments, we have observed that an appropriately chosen $\tau$ can achieve a high level of overlap (100%) while maintaining satisfactory accuracy results. We fully understand the concerns raised and will address them by refining the relevant statements in the paper during the rebuttal revision, aiming for a more balanced depiction of the scalability claims.
>
> **Q2: How to select $\tau$?**
>
> A2:
> Thank you for your question. We use grid search to select $\tau$. Specifically, we employ a simple grid search strategy within the range of {1, 3, 6, 12, 24, 48, 96, 192} to determine the value of $\tau$, aiming to achieve a balance between accuracy and throughput. To reduce computational demands during hyperparameter tuning for large models, we first conduct experiments on smaller model versions and then apply the hyperparameter to larger models. Similar to CO2, other methods such as Local-SGD/Adamw and SlowMo also need the tuning of $\tau$. In our practical implementation, we first fine-tune $\tau$ for CO2 and subsequently apply the same tuned $\tau$ values for Local-SGD/Adamw and SlowMo. Standardizing $\tau$ values across different methods for a given task ensures a fair and meaningful comparison of their convergence behavior and training throughput.
>
> Although grid search is used at present, there are more potential efficient strategies. One innovative suggestion is to perform a pre-run warmup with $\tau=1$ for a limited number of iterations. During this warmup phase, metrics like computation time and all-reduce communication time are collected to calculate an optimal number of steps for local computation ($\tau^*$). Then in formal runs, $\tau$ can be initialized as $\tau^*$, and adjusted based on accuracy results. Besides, varying $\tau$ values at different stages of training, similar to learning rates, is also an interesting idea. These ideas could be explored in future research related to CO2, we have not deeply investigated them currently.
>
> **Q3: How does the paper improve upon prior works which tune the communication frequency (see above for some references)?**
>
> A3:
> We thank for the insightful comments. The prior studies referenced above introduce adaptive communication frequency (i.e., the value of $\tau$) schedules on the native Local-SGD method. As for our work, the proposed CO2 leverages more sophisticated asynchronous communication design and outer updates on the top of naive Local-SGD, and further proposes staleness gap penalty and momentum clipping to improve convergence and training stability.
>
> We acknowledge that these adaptive schedules for $\tau$ are likely to yield a more favorable trade-off compared with using a constant $\tau$. However, these adaptive schedules can be seamlessly integrated into our CO2 algorithm. Exploring and incorporating such adaptive strategies is a promising direction for enhancing CO2 in the future research.

---

> ### Author Response · Authors · 2023-11-22
> **Response to Reviewer vfwG (Part 2)**
>
> **Q4: Implementation details and discussions on memory overheads**
>
> A4:
> Thank you for bringing this to our attention. We acknowledge the lack of details in our paper regarding the implementation of asynchronous communication and the analysis of memory overheads. In our implementation, the asynchronous communication operations are intentionally conducted on a new CUDA stream dedicated solely to communication. At the end of the inner loop, a collective communication primitive of NCCL all-reduce, is launched among all ranks within the communication group. Subsequently, a waiting operation is executed to ensure the completion of the last launched all-reduce from the preceding outer loop, allowing the subsequent computation operations to proceed.
>
> To facilitate asynchronous communication, a separate weight buffer is essential to store the stale version of the model. While this results in increased memory footprints compared to the synchronous version, it enables communication to overlap with multiple local steps. Specifically, in comparison to the SlowMo algorithm, CO2 incurs an additional memory footprint, twice that of the model itself, to support asynchronous communication and outer updates. To address redundant memory footprint concerns, techniques such as ZeRO-1, ZeRO-2, and ZeRO-3 have been integrated, alleviating issues related to optimizer states, gradients, and model parameters within the inner loop training of CO2. These measures aim to mitigate the negative impact of CO2's heightened memory overheads. Ongoing efforts are also dedicated to refining the CO2 algorithm in subsequent versions, with a specific focus on minimizing additional memory overheads.
>
> **Q5: Add the observed variances**
>
> A5:
> Thank you for bringing this to our attention. We have organized the experimental outcomes and presented the variance results on accuracy and throughput below. In terms of convergence results, Local-SGD/Adamw consistently exhibits the largest variances, due to lack of compensation measures for accuracy. SlowMo and CO2 demonstrate only slightly larger variances than SGD/Adamw, as they both integrate an outer loop to enhance convergence. Between CO2 and SlowMo, CO2 exhibits smaller variances due to the application of staleness gap penalty and momentum clipping.
>
> Regarding throughput, it is evident that all methods experience larger variances on larger clusters. This is reasonable as communication on larger clusters tends to be more unstable and fluctuates frequently. On large clusters, such as those with 64 or 128 GPUs, CO2 with different settings of $\tau$ demonstrates slightly larger variances than the baseline Adamw, regardless of the speed of inter-node networks.
>
> | Task | Model | SGD (Adamw) | Local-SGD (Local-Adamw) | SlowMo | CO2 |
> |-|-|-|-|-|-|
> | IC (A100)    | ResNet-50   | 76.92 ($\pm$ 0.05) | 75.57 ($\pm$ 0.76) | 77.12 ($\pm$ 0.11)  | 77.14 ($\pm$ 0.09) |
> | IC (A100)   | ViT (Base)  | 81.33 ($\pm$ 0.04) | 78.43 ($\pm$ 0.22) | 79.83 ($\pm$ 0.16)  | 80.95 ($\pm$ 0.08) |
> | IC (A100)    | VVT (Large) | 83.64 ($\pm$ 0.06) | 81.09 ($\pm$ 1.15) | 82.75 ($\pm$ 0.27) | 83.38 ($\pm$ 0.06) |
> | SS (3090)   | VVT (Large) | 47.82 ($\pm$ 0.05) | 44.25 ($\pm$ 2.24) | 47.51 ($\pm$ 0.12) | 47.80 ($\pm$ 0.11) |
> | PC (3090)   | Point-MAE   | 68.56 ($\pm$ 0.08) | 64.25 ($\pm$ 2.11) | 68.69 ($\pm$ 0.32) | 68.89 ($\pm$ 0.39) |
>
> | Task | Model   | Adamw  | Local-Adamw | SlowMo | CO2 |
> |-|-|-|-|-|-|
> | ALM (A100)  | GPT-2 (Small)   | 7.44 ($\pm$ 0.36)   | 7.95 ($\pm$ 2.04)    | 7.34 ($\pm$ 0.89)       | 7.37 ($\pm$ 0.73)       |
> | ALM (A100)  | GPT-2 (Medium)  | 6.61 ($\pm$ 0.53)   | 7.49 ($\pm$ 1.87)   | 6.41 ($\pm$ 0.65)       | 6.36 ($\pm$ 0.66)       |
> | ALM (A100)  | GPT-2 (Large)   | 6.26 ($\pm$ 0.58)   | 7.00 ($\pm$ 1.91)   | 6.29 ($\pm$ 0.61)       | 6.13 ($\pm$ 0.52)      |
> | ALM (A100)  | TN-LLM (7B)     | 16.82 ($\pm$ 0.86)  | 18.63 ($\pm$ 3.13)   | 16.95 ($\pm$ 1.01)      | 16.78 ($\pm$ 0.95)      |
> | BLM (3090)  | RoBERTa (Large) | 3.96 ($\pm$ 0.37)   | 4.38 ($\pm$ 0.83)    | 3.98  ($\pm$ 0.85)      | 3.95 ($\pm$ 0.96)       |
>
> | Ethernet | Method | Throughput | | | | |
> |-|-|-|-|-|-|-|
> |      |       | 8 GPUs     | 16 GPUs   | 32 GPUs | 64 GPUs | 128 GPUs |
> | RDMA  | CO2 ($\tau$=12) | 17980 ($\pm$ 48)     | 35692 ($\pm$ 126)              | 72491 ($\pm$ 183)    | 150073 ($\pm$ 362)   | 307557 ($\pm$ 617)    |
> | RDMA  | Adamw      | 19276 ($\pm$ 59)       | 38888 ($\pm$ 118)              | 77782 ($\pm$ 93)    | 151554 ($\pm$ 209)   | 289106 ($\pm$ 423)    |
> | TCP/IP | CO2 ($\tau$=48) | 18090 ($\pm$ 95)       | 35969 ($\pm$ 193)              | 71249 ($\pm$ 179)    | 147507 ($\pm$ 315)   | 304736 ($\pm$ 729)    |
> | TCP/IP  | CO2 ($\tau$=12) | 17995 ($\pm$ 72)       | 18975 ($\pm$ 108)              | 36095 ($\pm$ 151)    | 70839 ($\pm$ 373)    | 129865 ($\pm$ 564)    |
> | TCP/IP  | Adamw     | 18444 ($\pm$ 49)       | 3488 ($\pm$ 115)               | 7077 ($\pm$ 127)     | 12855 ($\pm$ 308)    | 22810 ($\pm$ 526)     |

---

> ### Author Response · Authors · 2023-11-22
> **Response to Reviewer vfwG (Part 3)**
>
> **Q6: How do other models considered in the paper (e.g., ResNets or ViTs) scale?**
>
> A6:
> Thank you for your inquiry. We specifically focused on assessing the scalability of TransNormer-LLM (7B) because the other models examined in our study are relatively small, with the number of parameters for each being smaller than 1B. As you rightly pointed out, smaller models tend to derive less benefit from communication optimizations. The communication overheads associated with all-reducing these small models are minimal, rendering the communication tails often negligible. Additionally, for computer vision (CV) task models such as ResNet or ViT, we typically do not scale them to very large sizes, although there is ongoing research exploring the scalability of ViT to 22B.
>
> To effectively demonstrate the scalability advantages of CO2, we conducted experiments specifically on GPT-3 (13B), a well-known language model with autoregressive transformer architecture. The results are shown in the below Table, includes throughput comparisons with SlowMo to address Q7. The results reveal that on GPT-3 (13B), CO2 consistently achieves higher throughput on platforms with different communication conditions.
>
> | Ethernet                | Method          | $\tau$ | Throughput | | | | |
> |-|-|-|-|-|-|-|-|
> |         |    |       |8 GPUs | 16 GPUs      | 32 GPUs | 64 GPUs | 128 GPUs |
> | RDMA   | CO2    | 12 |9468   |18587         |37905    |75682   |151846    |
> | RDMA   | SlowMo | 12 |9463   |18620         |37467    |75503   |148987    |
> | RDMA   | Adamw  |    |9519   |19131         |38824    |75326   |147723    |
> | TCP/IP | CO2    | 48 |9508   |18672         |38151    |75573   |151678    |
> | TCP/IP | SlowMo | 48 |9424   |18511         |38092    |75209   |146024    |
> | TCP/IP | CO2    | 12 |9324   |9694          |18792    |36968    |78638    |
> | TCP/IP | SlowMo | 12 |9359   |9628          |18085    |33864    |66908    |
> | TCP/IP | Adamw  |    |9482   |2283          |4192     |7984   |13832     |
>
> **Q7: How do other communication-efficient (e.g., SlowMo) methods scale on the fast and slow network?**
>
> A7:
> Thank you for your valid concerns. While Local-SGD/Adamw generally demonstrates commendable scalability performance, our investigation revealed that their convergence behaviors often fall short, as outlined in Table 1 and Table 2 in the paper. So we choose to only provide the detailed comparison between CO2 and SlowMo, considering their performance on both fast and slow networks. The experiments are conducted on GPT-3 (13B), are summarized in the Table corresponding to Q6. The results indicate that CO2 exhibits moderate advantages in terms of throughput compared to the communication-efficient method, SlowMo. Notably, these observed advantages will be more pronounced with larger clusters.
>
> **Q8: How much communication/computation overlap is actually achieved by CO2, particularly at scale?**
>
> A8:
> Thank you for the valuable feedback. Evaluating the communication/computation overlap is indeed crucial to substantiate the core claims of our paper. Notably, Table 3 in the manuscript provides a comprehensive overview of the scalability ratio of CO2 with varying $\tau$ across distinct communication platforms, offering a comparative analysis against Adamw. This scalability ratio can be considered as a metric for assessing the communication/computation overlap.
>
> To enhance our analysis, we conducted quantitative measurements on the time allocated to local computations and asynchronous communication. This assessment specifically targeted CO2 with different $\tau$ configurations on an A100 cluster equipped with 128 GPUs, employing a slower TCP/IP inter-node network. Adhering to the settings outlined in the paper, utilizing TransNormer-LLM (7B), we maintained consistency in our experimental conditions. The measured duration for a single-step local computation was approximately 0.109s, while an all-reduce communication incurred a cost of 1.566s. Subsequently, we computed and calculated the communication/computation overlap ratio for various $\tau$ values, presenting the results in the Table below.
>
> | Ethernet | Method | $\tau$ | Overlap Ratio|
> |-|-|-|-|
> | TCP/IP | CO2 | 48 | 100%   |
> | TCP/IP | CO2 | 24 | 100%   |
> | TCP/IP | CO2 | 12 | 83.28%   |
> | TCP/IP | CO2 | 6 | 41.81%   |
> | TCP/IP | CO2 | 3 | 20.39%   |
> | TCP/IP | CO2 | 1 | 6.52%   |
>
> **Q9: On the gradient bucketing**
>
> A9:
> I appreciate the reviewer's observation, and I agree with the noted concern. The depiction of gradient bucketing in the paper requires refinement, and we will address this in the revised version. It's important to clarify that the primary objective of gradient bucketing is to reduce the number of communication operations during gradient synchronization by employing larger tensor buffers in each operation. Communication/computation overlap is another system optimization build upon gradient bucketing.

---

> > ### Comment · Reviewer_vfwG · 2023-11-22
> > **Response**
> >
> > Thank you for the detailed and extensive response. This has addressed a number of my concerns and I have consequently raised my score.
> >
> > I hope to see these updates incorporated into the paper. In particular, the inclusion of variances, a more thorough discussion of choosing $\tau$, and the achieved communication/computation overlap are especially appreciated.

---

### Official Review · Reviewer_cSfz · 2023-11-09

**Soundness:** 3 good
**Presentation:** 3 good
**Contribution:** 2 fair
**Rating:** 6
**Confidence:** 4

**Summary:**

This work proposes a new distributed training algorithm called CO2, which aims to improve the communication efficiency of data-parallel training by overlapping local training iterations with parameter averaging from the previous global step. The proposed method is tested across multiple machine learning tasks and achieves better scalability than the baseline approaches while maintaining comparable convergence properties.

---
Post-rebuttal update: after reading authors' responses and other reviews, I decided to keep my weakly positive score unchanged and increase the confidence of my review. I think that the contributions of the study are solid and I am in favor of accepting the submission, but I am not fully sure that the work will have significant impact on the field in light of prior closely related publications.

**Strengths:**

* Overall, the proposed method is conceptually simple yet shows promising results.
* The paper has a broad range of experiments, covering 5 setups with models that are widely used in practice.
* Authors conduct a detailed ablation study for the components of CO2, as well as measure its scalability in different environments.

**Weaknesses:**

* While I am not an expert in distributed optimization, to my understanding, similar methods allowing full overlap of communication and computation have been proposed previously. See, for example, [1] from the related work section: on page 17, they state that "as long as the number of local updates τ is large enough, the communication can be fully overlapped with the local computation." This appears to be quite close to the primary contribution of this work, therefore I believe that the submission needs to describe the key distinctions from prior work in more detail.
* I think that the experimental setup description could benefit from more details. For example, while the authors mention that their hyperparameters were tuned "to find the optimal balance between efficiency and performance", we do not see neither the exact values of $\tau$ for each experiment nor the exact description of the tuning procedure. Also, authors mention that they leverage ZeRO for TransNormer experiments, but do not state the exact type of the optimizer within that family.
* Lastly, the majority of model sizes used in this work have quite small parameter counts (fewer than 1B), and therefore it is a bit surprising to see communication as the bottleneck for training even on 80Gbps networks. I think that it would be beneficial to provide more detailed breakdowns of computation and communication times (for example, the time to process 1 microbatch and 1 batch of data, as well as the time to exchange parameters) in each setting to demonstrate the necessity of large $\tau$.

[1] Cooperative SGD: A unified Framework for the Design and Analysis of Communication-Efficient SGD Algorithms. Jianyu Wang, Gauri Joshi. JMLR 2021

**Questions:**

* What were the values of $\tau$ for each experiment?
* Which stage of ZeRO have you used for the TransNormer experiment?
* In Table 2, it is somewhat surprising to see that CO2 (an asynchronous method) obtains consistently lower perplexity than a non-asynchronous adaptive method (AdamW). Do you have any explanations of that phenomenon?

---

> ### Author Response · Authors · 2023-11-22
> **Response to Reviewer cSfz (Part 1)**
>
> Dear reviewer,
>
> Thank you for your valuable and constructive feedback, which contributes a lot to the enhancement of our paper. Below, we provide our responses addressing your concerns. And all the revisions will be updated in the forthcoming rebuttal revision version of the paper.
>
> **Q1: Describe the key distinctions from prior work in more detail**
>
> A1:
> The main contribution of the Cooperative SGD paper is to unify a variety of local-update SGD algorithms, such as local SGD, elastic averaging SGD, and decentralized parallel SGD, under the proposed unified Cooperative SGD framework. It also provides a unified convergence analysis for the methods under this framework. As one of the by-products, it proposes to add an auxiliary variable $z$ on the elastic averaging SGD, which serves as a local copy of the model. It can be seen as a preliminary approach that enables the asynchronous communication of model averaging to overlap with computation. However, our method leverages more sophisticated asynchronous communication design to improve the convergence. The differences between our method and the Cooperative SGD are threefold: 1) The CO2 algorithm is built upon SlowMo, which involves slow momentum and outer updates to improve the convergence; 2) CO2 leverages more sophisticated asynchronous communication design together with the outer momentum and outer updates to hide more communications; 3) CO2 also introduces staleness gap penalty and outer momentum clipping to improve the convergence as well as training stability.
>
>
> **Q2: The values of $\tau$ for each experiment and the description of the tuning procedure**
>
> A2:
> Thanks for the valuable comments. We agree that describing the tuning procedure and the setting of $\tau$ in experiments are helpful for understanding the algorithm. In our experiments, we tune the values of $\tau$ for each task using a simple grid search strategy within the range of {1, 3, 6, 12, 24, 48, 96, 192} to reach a balance of accuracy and throughput. The tuning experiments are conducted on smaller versions of models when the model to be trained is large, owing to the associated high time and resource costs. We start the tuning of $\tau$ from 1 to larger candidate values given the consideration of high-speed communication on the corresponding platform. It is worth noting that, in addition to CO2, other methods such as Local-SGD/Adamw and SlowMo also require tuning of $\tau$. In practice, we exclusively tune $\tau$ for CO2 and employ the same tuned $\tau$ values for Local-SGD/Adamw and SlowMo. This is because of the similar role played by $\tau$ in these local-updating methods. Employing identical $\tau$ values for different methods on the same task ensures a fair comparison of their convergence and training throughput.
>
> We list the values of $\tau$ in all the convergence experiments in the paper as below:
>
> | Task | Model| SGD (Adamw) | Local-SGD (Adamw) | SlowMo     | CO2 |
> |-|-|-|-|-|-|
> |      |             | Acc  | Acc / $\tau$             | Acc / $\tau$ | Acc / $\tau$ |
> | IC (A100)    | ResNet-50   | 76.92      | 75.57 / 1                  | 77.12 / 1       | 77.14 / 1       |
> | IC (A100)   | ViT (Base)  | 81.33        | 78.43 / 3                   | 79.83 / 3       | 80.95 / 3       |
> | IC (A100)    | VVT (Large) | 83.64        | 81.09 / 1                   | 82.75 / 1       | 83.38 / 1       |
> | SS (3090)   | VVT (Large) | 47.82        | 44.25 / 6                   | 47.51 / 6       | 47.80 / 6       |
> | PC (3090)   | Point-MAE   | 68.56        | 64.25 / 3                   | 68.69 / 3       | 68.89 / 3       |
>
> | Task | Model| Adamw      | Local-Adamw | SlowMo     | CO2|
> |-|-|-|-|-|-|
> |      |                 | Acc | Acc / $\tau$  | Acc / $\tau$ | Acc / $\tau$ |
> | ALM (A100)  | GPT-2 (Small)   | 7.44       | 7.95 / 3        | 7.34 / 3       | 7.37 / 3       |
> | ALM (A100)  | GPT-2 (Medium)  | 6.61       | 7.49 / 3       | 6.41 / 3       | 6.36 / 3       |
> | ALM (A100)  | GPT-2 (Large)   | 6.26       | 7.00 / 6       | 6.29 / 6       | 6.13 / 6      |
> | ALM (A100)  | TN-LLM (7B)     | 16.82      | 18.63 / 12       | 16.95 / 12      | 16.78 / 12      |
> | BLM (3090)  | RoBERTa (Large) | 3.96       | 4.38 / 6        | 3.98 / 6       | 3.95 / 6       |
>
> **Q3: Which stage of ZeRO have you used for the TransNormer experiment?**
>
> A3:
> Thanks for the question. We use ZeRO stage 2 in the experiments of TransNormer-LLM, will add this information into the paper later.

---

> ### Author Response · Authors · 2023-11-22
> **Response to Reviewer cSfz (Part 2)**
>
> **Q4: Detailed breakdowns of computation and communication times**
>
> A4:
> Thank you for the valuable comments. We conducted quantitative measurements on the time allocated to local computations and asynchronous communication. This assessment specifically targeted CO2 with different $\tau$ configurations on an A100 cluster equipped with 128 GPUs, employing a slower TCP/IP inter-node network. Adhering to the settings outlined in the paper, utilizing TransNormer-LLM (7B), we maintained consistency in our experimental conditions. The measured duration for a single-step local computation was approximately 0.109s, while an all-reduce communication incurred a cost of 1.566s. Subsequently, we computed and calculated the communication/computation overlap ratio for various $\tau$ values, presenting the results in the Table below.
>
> | Ethernet | Method | $\tau$ | Overlap Ratio|
> |-|-|-|-|
> | TCP/IP | CO2 | 48 | 100%   |
> | TCP/IP | CO2 | 24 | 100%   |
> | TCP/IP | CO2 | 12 | 83.28%   |
> | TCP/IP | CO2 | 6 | 41.81%   |
> | TCP/IP | CO2 | 3 | 20.39%   |
> | TCP/IP | CO2 | 1 | 6.52%   |
>
>
>
> **Q5: Why CO2 obtains consistently lower perplexity than AdamW?**
>
> A5:
> The elements that have an impact on accuracy in CO2 are multiple sides: asynchronous operation and local updating may hurt the convergence as them both introduce noise into the training procedure, while staleness gap penalty and outer momentum clipping may improve the convergence. The resulted accuracy is a mixture of all these elements. Furthermore, in our experiments, we found that in some cases, moderate values of $\tau$ (i.e., the number of local steps) have a regularizing effect, which leads to an improvement on the accuracy results.

---

### Official Review · Reviewer_8oQm · 2023-11-11

**Soundness:** 3 good
**Presentation:** 4 excellent
**Contribution:** 3 good
**Rating:** 8
**Confidence:** 4

**Summary:**

This paper proposed a novel distributed training method: CO2, which can overlap communication and computation in distributed training. This technique is particularly useful when there are a large number of GPU nodes and the inter-connection between nodes are very slow. Compared to previous works, this paper introduces (1) penalty on stale momentum (2) momentum clipping. Empirical ablations show these two techniques are crucial to improve the training convergence performance. The authors also conducted extensive empirical studies, including experiments on image classification, large language model training, to demonstrate the effectiveness of the proposed method.

**Strengths:**

- The paper has extensive empirical studies across different learning tasks as well as different network environments.
- The authors also provided a convergence analysis for the proposed method.

**Weaknesses:**

- The idea of overlapping communication and computation is not new, as mentioned in the paper. The key contribution of this paper would be introducing the staleness penalty and momentum clipping mechanisms. They also present solid experimental resutls.
- The comparison with previous works are not enough. For example, totally overlapping communication and computation has already been achieved. like Wang et al, 2020. Overlap-Local SGD. The authors should include more discussions on the differences. or even include this method as a baseline.
- It is not very clear the convergence analysis was performed on which algorithm. Does the analysis consider staleness penalty and momentum clipping? Also, the convergence analysis looks like following previous works. It'd be better to cite few at the very beginning of the analyses.

**Questions:**

See the above section.

---

> ### Author Response · Authors · 2023-11-22
> **Response to Reviewer 8oQm**
>
> Dear reviewer,
>
> We thank for the reviewer's comprehensive evaluation on our paper. Below, we provide our responses addressing your concerns. And all the revisions will be updated in the forthcoming rebuttal revision version of the paper.
>
> **Q1: The idea of overlapping communication and computation is not new. The comparison with previous works are not enough, like Wang et al, 2020. Overlap-Local SGD**
>
> A1:
> We thank for the valuable comments. We will include the Overlap-Local SGD method in our comparisons. The method is also able to overlap communication by local computations via adding an anchor model on each node. The differences between our method and the Overlap-Local SGD are threefold:
> 1) Overlap-Local SGD achieves communication-computation overlap on the top of naive Local-SGD, whose convergence performance is normally not good enough. On the other hand, the CO2 is built on SlowMo, which involves outer loop includes outer momentum and outer updates to improve the convergence of native Local-SGD.
> 2) As mentioned by the reviewer, CO2 introduced staleness gap penalty and outer momentum clipping to prevent divergence and improve training stability, which has been verified effective via extensive experiments.
> 3) The Overlap-Local SGD is only tested on the CIFAR-10 image classification task using ResNet-18, while CO2 has been widely verified on multiple tasks on both CV and NLP fields, including image classification, semantic segmentation, 3D point cloud reconstruction, autoregressive language modeling and bidirectional language modeling.
>
> To better comparing the performance of Overlap-Local SGD and CO2, we reproduced Overlap-Local SGD on its open-source implementation and ported it into our codebase. We run the experiments in paper using Overlap-Local SGD and present the results as below. We also list the throughput results and the settings of $\tau$ in all the convergence experiments we have conducted.
>
> | Task | Model| SGD (Adamw) | Local-SGD(Adamw) | Overlap-Local-SGD(Adamw) | SlowMo     | CO2 |
> |-|-|-|-|-|-|-|
> |      |             | Acc / Thpt  | Acc / Thpt / $\tau$ | Acc / Thpt / $\tau$ | Acc / Thpt / $\tau$ | Acc / Thpt / $\tau$ |
> | IC (A100)    | ResNet-50   | 76.92 / 108739      | 75.57 / 108758 / 1 | 76.28 / 108765 /1 | 77.12 / 108741 / 1       | 77.14 / 108753 / 1       |
> | IC (A100)   | ViT (Base)  | 81.33 / 39422        | 78.43 / 39512 / 3 | 78.04 / 39511 /3 | 79.83 / 39509 / 3       | 80.95 / 39533 / 3       |
> | IC (A100)    | VVT (Large) | 83.64 / 44375        | 81.09 / 44390 / 1  | 80.33 / 44392 /1 | 82.75 / 44376 / 1       | 83.38 / 44387 / 1       |
> | SS (3090)   | VVT (Large) | 47.82 / 5384        | 44.25 / 5528 / 6   | 45.21 / 5545 /6 | 47.51 / 5521 / 6       | 47.80 / 5562 / 6       |
> | PC (3090)   | Point-MAE   | 68.56 / 5859        | 64.25 / 5931 / 3   | 63.78 / 5950 /3 | 68.69 / 5904 / 3       | 68.89 / 5956 / 3       |
>
> \* For IC and SS tasks, we present the throughput results in "images/s", the image resolution is 224x224; For PC task, we present the throughput results in "point clouds/s", each point cloud has 1024 points.
>
> | Task | Model| Adamw      | Local-Adamw | Overlap-Local-SGD(Adamw) | SlowMo     | CO2|
> |-|-|-|-|-|-|-|
> |      |                 | Acc / Thpt | Acc / Thpt / $\tau$  | Acc / Thpt / $\tau$ | Acc / Thpt / $\tau$ | Acc / Thpt / $\tau$ |
> | ALM (A100)  | GPT-2 (Small)   | 7.44 / 6.543e6       | 7.95 / 6.556e6 / 3        | 8.11 / 6.556e6 / 3 | 7.34 / 6.554e6 / 3       | 7.37 / 6.556e6 / 3       |
> | ALM (A100)  | GPT-2 (Medium)  | 6.61 / 2.084e6       | 7.49 / 2.094e6 / 3       | 7.26 / 2.092e6 / 3 | 6.41 / 2.091e6 / 3       | 6.36 / 2.092e6 / 3       |
> | ALM (A100)  | GPT-2 (Large)   | 6.26 / 1.052e6       | 7.00 / 1.059e6 / 6       | 7.18 / 1.057e6 / 6 | 6.29 / 1.053e6 / 6       | 6.13 / 1.056e6 / 6      |
> | ALM (A100)  | TN-LLM (7B)     | 16.82 / 0.281e6      | 18.63 / 0.303e6 / 12       | 17.83 / 0.306e6 / 12 | 16.95 / 0.301e6 / 12      | 16.78 / 0.308e6 / 12      |
> | BLM (3090)  | RoBERTa (Large) | 3.96 / 2262       | 4.38 / 2815 / 6        | 4.52 / 2877 / 6 | 3.98 / 2794 / 6       | 3.95 / 2892 / 6       |
>
> \* For ALM and BLM tasks, we present the throughput results in "tokens/s".
>
> **Q2: About the convergence analysis**
>
> A2:
> We thank for the comprehensive review. The convergence analysis was performed on CO2 with Local-SGD used in the inner loop. Staleness gap penalty and momentum clipping are not yet considered in the current convergence analysis. This is because we take these two techniques as plug-ins, a convergence analysis without them is more clean and easy to follow. The proof derivation in SlowMo is taken as a reference when we do convergence analysis for CO2, we will follow the suggestions to add the citation of it in the beginning of the proof.

---

### Meta-Review · Area_Chair_P8jK · 2023-12-07

**Metareview:**

This paper introduces a new approach, CO2, to distributed training which incorporates ideas of asynchronous updating, combined with a staleness penalty and momentum clipping which alleviate negative effects due to staleness. An extensive empirical evaluation verifies the promise of the approach and illustrates its utility.

Reviewers raised some concerns but on the whole were very positive about this paper. In particular, the thorough evaluations and clear motivation for the work make the paper very strong. The authors also improved the paper in response to reviewer comments, taming down claims about scaling behavior and clarifying the importance of tuning $\tau$ properly, among other points.

The paper makes a very strong contribution to the literature on distributed training and is likely to prove useful to practitioners.

**Justification For Why Not Higher Score:**

I wouldn't be opposed to having this as an oral.
The main reasons I think a spotlight is appropriate is that the core ideas of the approach are largely based on prior work (e.g., SlowMo). There is clearly novelty in getting this to work efficiently and to scale well, and the thorough empirical evaluation is very convincing. That could be a reason to bump it up. That said, the details of how they get it to work (penalizing staleness, clipping momentum) may not be of broad interest to the entire NeurIPS community.

**Justification For Why Not Lower Score:**

I think this is a very solid contribution. Moreover, distributed training for neural networks has been a very well-investigated area, and it's not as common for there to be seriously novel advances. I consider this to be substantial enough to justify at least a spotlight, so I would argue strongly against only accepting this as a poster.

---

### Decision · Program_Chairs · 2024-01-16

Accept (spotlight)